# Structural basis of topoisomerase targeting by delafloxacin

Shabir Najmudin[1,5], Xiao-Su Pan[1,5], Beijia Wang [2,5], Lata Govada[2], Naomi E. Chayen [2], Noelia Rubio[3], Milo S. P. Shaffer [3], Henry S. Rzepa[4], L. Mark Fisher [1] ✉ & Mark R. Sanderson [1,2] ✉

Delafloxacin is a potent anionic fluoroquinolone approved for the treatment of respiratory infections that acts by trapping the DNA cleavage complexes of bacterial topoisomerase IV and gyrase. Its N-1-pyridinyl-, C-7-azetidinyl- and C-8-chlorine substituents confer enhanced antibiotic activity against bacteria resistant to other fluoroquinolones, but its mode of action is unclear. Here we present the X-ray crystal structures of a delafloxacin-DNA cleavage complex obtained by co-crystallization with *Streptococcus pneumoniae* topo IV using a graphene nucleant and solved at 2.0 and 2.4 Å resolution. The two $Mg^{2+}$-chelated delafloxacin molecules intercalated at the DNA cleavage site are bound in an unusual conformation involving interacting out-of-plane N-1-aromatic- and C-8-chlorine- substituents. The unprecedented resolution allows comprehensive imaging of water-metal ion links integrating enzyme and DNA through drug-bound and active-site $Mg^{2+}$ ions plus the discovery of enzyme-bound $K^+$ ions. Our studies on delafloxacin action suggest that intrinsic target affinity contributes to its activity against quinolone-resistant bacteria.

Fluoroquinolones are an important class of antimicrobial drugs that target topoisomerase (topo) IV and DNA gyrase, the type IIA topoisomerases (Top2As) essential for bacterial growth (Fig. 1 and Supplementary Fig. 1)[1,2]. The attractive properties of fluoroquinolones are their oral availability, clinical potency, and activity against a wide range of bacterial pathogens[1]. Given these features, the World Health Organisation has designated fluoroquinolones as one of five antimicrobial drug classes critical to the management of human disease[3]. Currently, ciprofloxacin, levofloxacin and moxifloxacin are the fluoroquinolones in widest clinical use and have recently been joined by delafloxacin, a novel drug with a chemically distinct structure (Fig. 1)[1].

Delafloxacin (ABT-492, Baxdela[R], Quofenix[R]) is a unique anionic fluoroquinolone with impressive activity against Gram-positive and Gram-negative pathogens as well as anaerobes[4]. The drug was approved in 2017 for the treatment of bacterial skin/skin structure infections later extended to community acquired pneumonia[4].

*S. pneumoniae*, also known as the pneumococcus, is the main cause of community acquired pneumonia and is also an important pathogen in meningitis and sepsis[4]. Pneumococcal pneumonia affects the very young and old with an estimated one million deaths worldwide in children under five and the problem of drug-resistant disease[5,6].

Delafloxacin is a potent antibacterial agent that shares the 4-quinolone/naphthyridone core common to all fluoroquinolones but differs in three unusual structural features. First, it has a large heteroaromatic group at N-1 and a neighbouring C-8 chlorine that increase its activity in vitro (Fig. 1) and that re-establish efficacy against bacteria resistant to other therapeutic fluoroquinolones[4,7,8]. Second, the absence of a protonable substituent at C-7 means that the anionic drug molecule has a single charge at physiological pH, but is charge neutral at mildly acidic pH, promoting greater bacterial uptake/accumulation and enhanced bacteriological activity[4,9]. By contrast, the zwitterionic fluoroquinolones such as levofloxacin and moxifloxacin

[1]Molecular and Cellular Sciences Section, Neuroscience and Cell Biology Research Institute, City St George's, University of London, Cranmer Terrace, London, UK. [2]Division of Systems Medicine, Department of Metabolism, Digestion and Reproduction, Faculty of Medicine, Imperial College London, London, UK. [3]Departments of Chemistry & Materials Science, Imperial College London, London, UK. [4]Department of Chemistry, Imperial College London, London, UK. [5]These authors contributed equally: Shabir Najmudin, Xiao-Su Pan, Beijia Wang. ✉e-mail: lfisher@sgul.ac.uk; mark.sanderson@imperial.ac.uk

**Fig. 1 | Chemical structures of fluoroquinolones.** Structures of the clinically important fluoroquinolones delafloxacin, levofloxacin, moxifloxacin and ciprofloxacin are shown alongside clinafloxacin and trovafloxacin, two potent investigational agents.

are positively charged at acidic pH, are taken up less readily and exhibit decreased activity under acidic conditions[9]. The differential effects of the acidic conditions that prevail in the lung, bladder, skin, biofilms and sites of inflammation may in part explain the clinical efficacy of delafloxacin against bacteria resistant to levofloxacin, moxifloxacin and ciprofloxacin[4]. However, delafloxacin also displays strong activity in vitro against fluoroquinolone-resistant strains of *S. aureus* and *S. pneumoniae* bearing mutations in topo IV and gyrase which could suggest that intrinsic drug affinity for topo IV and gyrase also has a role in differential activity[4,8].

Topo IV and gyrase are ubiquitous ATP-dependent enzymes co-expressed in most bacterial species and that perform essential cellular activities (see ref 2 and references therein). Topo IV is a tetrameric complex made up of two ParE and two ParC subunits (Fig. 2) that mediates unlinking of interlocked daughter chromosomes[10,11]. By contrast, the closely homologous gyrase GyrA$_2$GyrB$_2$ tetramer controls DNA supercoiling and facilitates DNA replication[12,13]. Both enzymes alter DNA topology by passing a DNA helix through a transient double strand break in a second DNA segment (the 'G or gate' DNA) involving a covalent enzyme-DNA 'cleavage complex' (Fig. 2)[14,15]. Fluoroquinolones stabilise this complex which cellular processes convert into a lethal lesion, likely a frank double-stranded DNA break[1,2]. Fluoroquinolone resistance often arises from point mutations at two highly conserved serine and acidic residues in ParC/GyrA that reduce drug binding, often with both enzymes mutated in resistant clinical isolates[1,2,16–18].

Whereas previous work on delafloxacin has largely concentrated on *S. aureus* and MRSA in skin infections, we have focused our attention on *S. pneumoniae*, an important cause of systemic life threatening disease[4]. The development of anti-pneumococcal fluoroquinolones has been a significant advance but our understanding of delafloxacin activity and mechanism has been hindered by the lack of any structural information on how it engages its topoisomerase targets.

Previously, by co-crystallising a core complex of *S. pneumoniae* topo IV with the strongly cleaved pneumococcal E-site DNA (Fig. 2c) in the presence of clinafloxacin or moxifloxacin, we obtained the X-ray crystal structures of cleavage complexes stabilised by fluoroquinolones[19–21]. Subsequently, we and others have solved structures of E-site core cleavage complexes for topo IV and gyrase from different bacterial species and stabilised by different fluoroquinolones[22–24], and by other inhibitors that occupy the fluoroquinolone binding pocket such as quinazolinediones and spiropyrimidinetriones e.g., zoliflodacin[21,25–28]. Structures have also been obtained for the core cleavage complex stabilised by compounds that bind at distinct non-quinolone sites such as new bacterial topoisomerase inhibitors (NBTIs) e.g., gepotidacin, as well as allosteric thiophene inhibitors, and complex antibiotics e.g., evybactin that bind across two sites[29–32]. Zoliflodacin and gepotidacin have recently had a positive outcome in Phase 3 clinical trials and have been approved for the treatment of gonorrhoea[2]. To date the available X-ray crystal structures with fluoroquinolones are at medium-low resolution

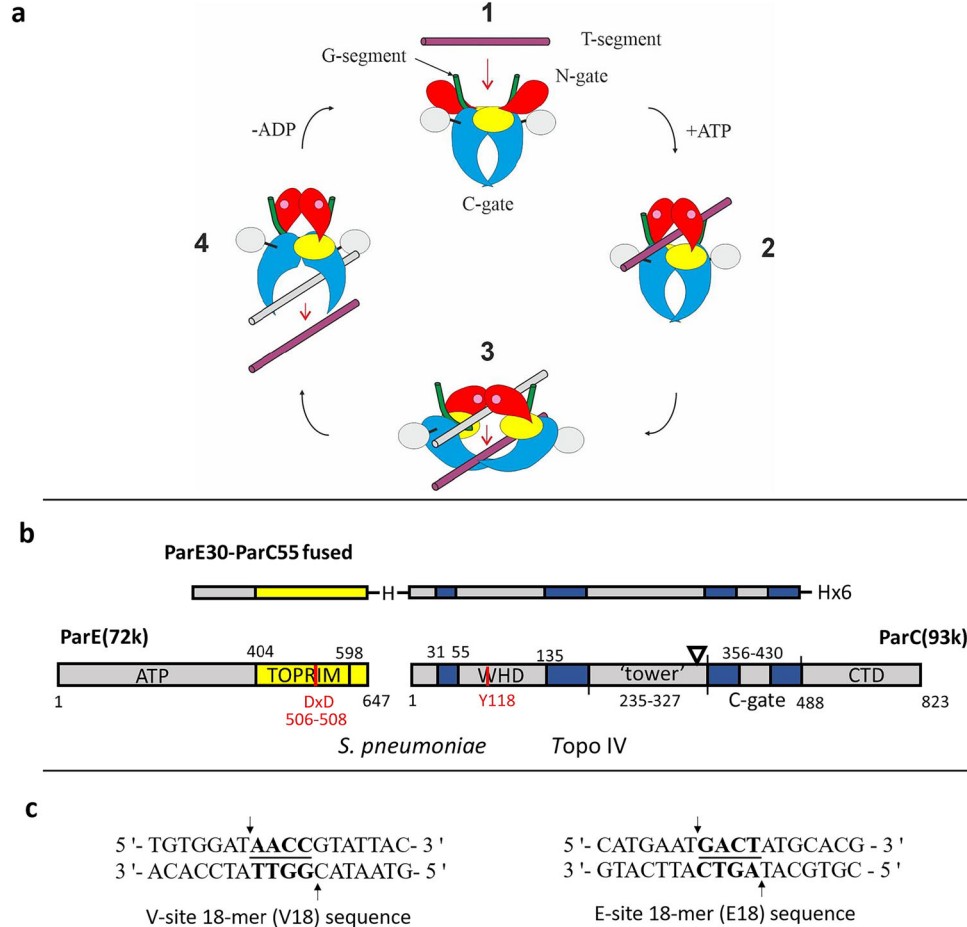

**Fig. 2 | Mechanism and structure of type IIA topoisomerases. a** Proposed catalytic cycle of a type II topoisomerase illustrated for topo IV. The transported T-segment DNA (see stage **1**) is captured by closure of the N-gate formed by ATP-induced dimerisation of the ParE ATPase domains (red) which allows its presentation to the cleavage core of the enzyme (**2**), passage through the G-segment DNA bound at the DNA cleavage gate (**3**) and subsequent release through the protein C-gate (**4**). Hydrolysis of bound ATP (pink dots) resets the enzyme for another cycle. The breakage-reunion (ParC55) and C-terminal domain (CTD) of ParC are in blue and silver; the TOPRIM domain of ParE (ParE30) is in yellow. G-gate DNA and transported (T) segment DNA are shown in green and purple, respectively.

Bound ATP is shown by pink circles (Reproduced with permission under a Creative Commons licence from Fig. 1 of Laponogov, I et al., Nucleic Acids Res, **41**, 9911-9923 (2013). Complexes **1**, **2** and **3** (either with or lacking the ATPase domains and CTDs as in the core complex) can be captured as a cleavage complex by fluoroquinolones. **b** Domain organisation for the fused ParE-ParC cleavage core of *S. pneumoniae* topo IV (top) shown with that of the corresponding full-length ParE and ParC subunits (bottom). The open triangle shows the location of a potassium ion binding motif. **c** 18-mer E-site (E18) and V-site (V18) DNA duplexes are strongly cleaved (arrows) by *S. pneumoniae* topo IV in the presence of quinolones generating a 4 bp overhang (underlined).

(-2.9–3.5 Å) except for the 2.4–2.6 Å E-site structures obtained with *M. tuberculosis* gyrase[33]. Structures at higher resolution have proven elusive but will be essential for a full understanding of fluoroquinolone action.

Here we present the X-ray crystal structures of a delafloxacin-DNA cleavage complex, that of *S. pneumoniae* topo IV, including an unprecedented 2.0 Å resolution structure obtained using a novel nucleant to induce crystallisation. In combination with biochemical and computational studies, our work provides new insights on how delafloxacin binds the cleavage complex to retain clinical activity against resistant strains, an important paradigm in designing drugs to counter antimicrobial resistance.

## Results

### Capture of topo IV and gyrase cleavage complexes

To facilitate crystallisation and structural work, we first identified the topoisomerase target with the higher affinity for delafloxacin by examining drug-induced DNA cleavage by *S. pneumoniae* topo IV and gyrase. Supercoiled plasmid pBR322 DNA was incubated with topo IV or gyrase in the absence or presence of increasing concentrations of delafloxacin. Following addition of SDS (to release DNA breaks) and

proteinase K (to remove bound protein), the DNA products were separated and analysed by agarose gel electrophoresis (Fig. 3a). For both topo IV and gyrase, delafloxacin promoted efficient single- and double-stranded breakage of the input DNA forming both nicked and linear DNA products. In fact, high levels of single-stranded breaks were seen, especially with topo IV. Higher drug concentrations produced a smear of smaller DNA fragments due to multiple double-strand cleavage events on each plasmid DNA (Fig. 3a). Judged either by the disappearance of the supercoiled DNA band or the formation of double-stranded DNA breaks, topo IV was trapped more efficiently, with 0.25 µM delafloxacin producing similar levels of DNA cleavage to gyrase at 2.5–5.0 µM drug. Based on our previous experience that high quinolone affinity favours crystallisation of quinolone cleavage complexes, this ~10- to 20-fold greater stabilisation of topo IV cleavage complexes compared with gyrase led us to focus our structural work on topo IV.

Attempts thus far to obtain high resolution X-ray crystal structures of quinolone-stabilised holoenzyme cleavage complexes have been unsuccessful. However, previously we established that a 'core cleavage complex' comprising two copies each of the metal binding ParE30 TOPRIM and ParC55 breakage-reunion domains retains all the

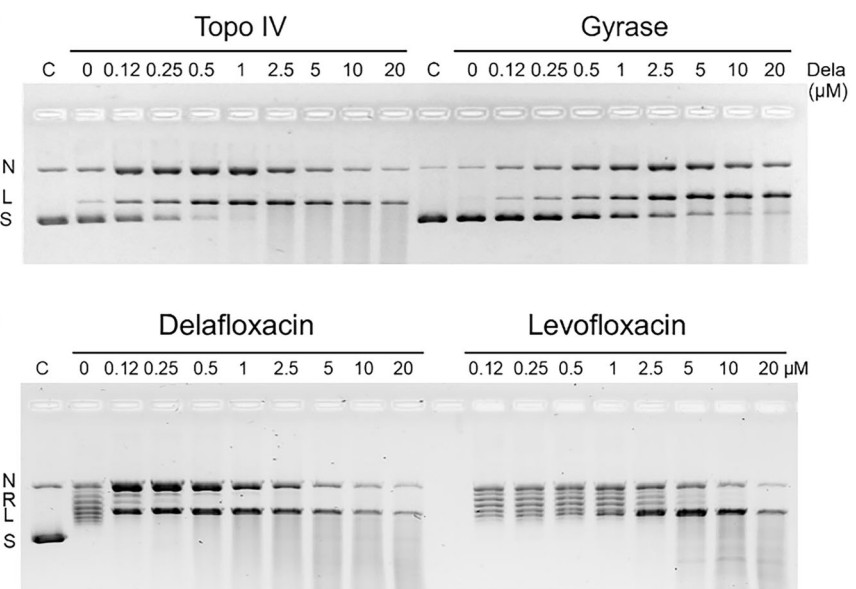

**Fig. 3 | Delafloxacin stabilises the cleavage complex of pneumococcal topo IV and gyrase. a**. Delafloxacin-mediated DNA cleavage is more efficient for *S. pneumoniae* topo IV than gyrase. Supercoiled pBR322 DNA (0.4 μg) was incubated in cleavage buffer (40 mM Tris·HCl (pH 7.5), 6 mM MgCl₂, 10 mM dithiothreitol, 200 mM potassium glutamate, 50 μg/ml bovine serum albumin) with either topo IV (reconstituted from 0.45 μg ParC and 1 μg ParE) or gyrase (0.45 μg GyrA/1 μg GyrB) in the absence or presence of delafloxacin (Dela) at the indicated concentrations. After incubation for 60 min at 37 °C, samples were treated with sodium dodecyl sulphate and proteinase K. Plasmid DNA products were separated by gel electrophoresis in 1% agarose run in TBE buffer. Subsequent gel staining with ethidium bromide allowed DNA visualisation and photography under UV light. Lanes C, supercoiled pBR322 DNA. S, L and N denote supercoiled, linear and nicked DNA products. **b** The ParE30-ParC55 fusion protein is active in DNA cleavage and

targeted more efficiently by delafloxacin than levofloxacin. The protein (0.4 μg) was incubated in cleavage buffer with supercoiled pBR322 DNA (0.4 μg) in the absence or presence of delafloxacin or levofloxacin at the concentrations indicated on the figure. Induction of cleavage and analysis of DNA cleavage products was carried out as described for Fig. 3a. N, R, L and S denote nicked, relaxed, linear and supercoiled DNA, respectively. Cleavage experiments were conducted once primarily to aid structural studies (rather than provide inhibition parameters to decimal place precision). Experiments were carefully done and show that as for other quinolones, delafloxacin captures a cleavage complex markedly more efficiently (10-20-fold) with topo IV than gyrase. Studies in Fig. 3b show delafloxacin is comparably active against the topo IV fusion and holoenzyme complexes and for levofloxacin recapitulate results we previously published using the same conditions and levels of drug, fusion protein and pBR322 DNA[23].

DNA cleavage functions of the holoenzyme[19,20]. To further stabilise the core complex for structural work, we generated a recombinant ParE30-ParC55 fusion protein (Fig. 2b) which was tested in the cleavage assay with delafloxacin and with levofloxacin (Fig. 3b). In the absence of drug, the fusion protein exhibited an ATP-independent relaxing activity that converted the supercoiled plasmid DNA into a ladder of relaxed DNA topoisomers (Fig. 3b), a previously known activity of this core complex[23]. The fusion protein was fully active in DNA cleavage, with delafloxacin again trapping higher levels of single-strand- over double-strand DNA breaks. By contrast, levofloxacin yielded predominantly double-stranded DNA breaks. Interestingly, delafloxacin was intrinsically some 20-fold more potent than levofloxacin with 0.12 μM delafloxacin inducing the same level of DNA cleavage as 2.5 μM levofloxacin (Fig. 3b). These levofloxacin cleavage results replicate the outcome of a previous study using the fusion protein under the same conditions, demonstrating consistency of approach[23].

**The delafloxacin-topo IV cleavage complex at 2.0 Å resolution**
Co-crystallisation trials were set up to include the ParE30-ParC55 fusion protein, the strongly cleaved 18-mer E-site DNA duplex (present on the pneumococcal chromosome) or the 18-mer V-site DNA sequence (present in ColE1 plasmids) (Fig. 2c) plus 2 mM delafloxacin to saturate drug binding[19,34]. Crystals of the cleavage complex were obtained with both DNA duplexes and the structures were solved at 2.4 Å resolution by molecular replacement (Supplementary Table 1). Moreover, a V18 structure solved at an unprecedented 2.0 Å resolution (Fig. 4, Supplementary Table 1) was obtained by improving the crystal quality through seeding with a graphene-PEG nucleant[35]. The PEG chains of the nucleant are thought to attract the complex to the flat and porous graphene surface, creating a localised supersaturated

environment favouring protein aggregation. Except for analysis of the drug chelated Mg²⁺ (which is better defined in the 2.4 Å E-site structure solved by combining multiple data sets), we focus largely on the 2.0 Å V-site structure, whose overall higher resolution meant that the electron densities of the bound drug, the base pairs of the DNA, the puckering of sugars of the DNA, and the side chains and main chain of the protein are generally much better defined as is the water structure. Moreover, we see >1000 individually imaged water molecules including those in the inner and outer co-ordination spheres of the two pairs of functionally distinct magnesium ions, and at a pair of K⁺ sites discovered here in ParC. These structures of a fluoroquinolone-cleavage complex are much better defined than those solved previously and consequently illuminate the role of metal ions and water molecules in topoisomerase catalysis, drug binding and fluoroquinolone resistance.

**Canonical features of the delafloxacin-stabilised complex**
The delafloxacin-topo IV-DNA cleavage complex exhibits both shared and highly distinctive features compared to complexes formed with other fluoroquinolones reported earlier[20,22–24]. The two halves of the enzyme structures are related by non-crystallographic symmetry, with the dimer as the asymmetric subunit. The overall architecture of the complex is described here with delafloxacin-specific aspects covered in following sections. The delafloxacin complex presents as a dimer with the two ParC55 breakage-reunion domains in a 'closed' conformation flanked by the two ParE30 metal binding TOPRIM domains, with a single DNA helix bound across the dimer interface (Fig. 4). The ParE domains are held close to the ParC dimer by the ParC α-1 arms (residues 6-30), an essential interaction for DNA cleavage and catalysis[23]. The ParE domains- formed of four parallel β-sheets and surrounding α-helices- plus the ParC winged helix domain (WHD) and

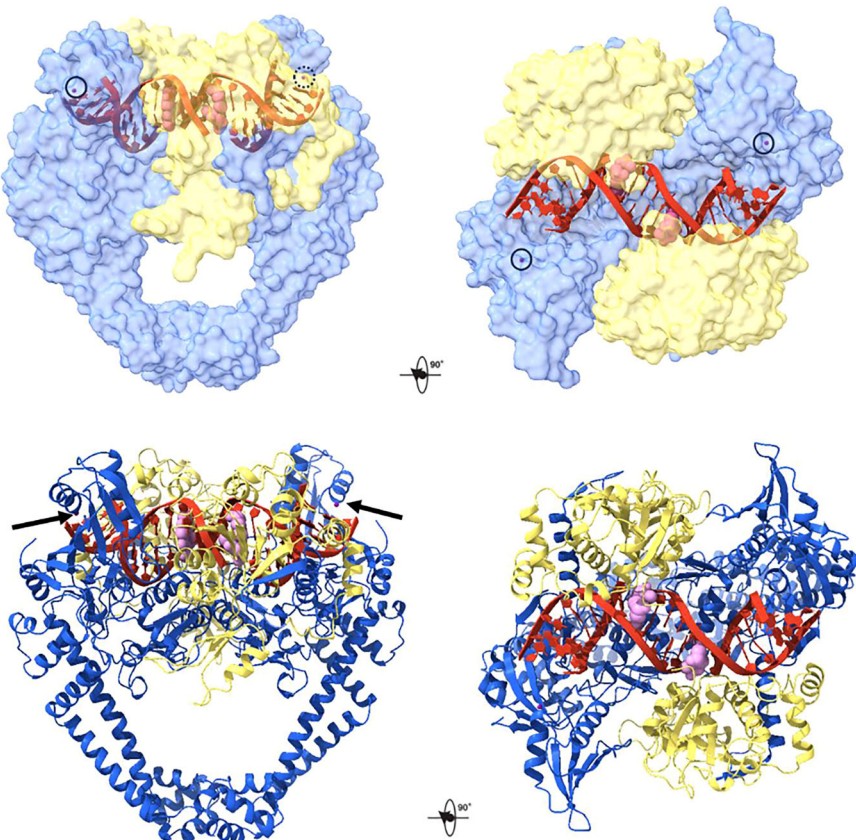

**Fig. 4 | Orthogonal views in surface representation of the 2.0 Å X-ray crystal structure of the delafloxacin-core cleavage complex of *S.pneumoniae* topo IV with 18-mer V gate DNA (PDB ID: 8QMB).** The ParE TOPRIM domains are in yellow, ParC domains in blue, DNA is in red (cartoon representation) and the two drug molecules are in pink (solid sphere representation). The two surface potassium ions bound one to each ParC tower are in dark purple (solid sphere representation) highlighted by dark circles (top) and arrows (bottom left).

'tower' domains together form the protein groove that accommodates a single V-site (or E site DNA) duplex as the G-gate DNA (Fig. 2b, and Supplementary Fig. 2 and Supplementary Movie 1). Protein binding induces a U-shaped bend in the DNA involving the symmetric minor groove intercalation of the two ParC I170 sidechains. The downstream C-gate region of ParC (Fig. 4) consists of a pair of two long α-helices terminated by a short α-helix, forming a 30 Å cavity that can accommodate the transported DNA prior to passage and release through the protein C-gate (Fig. 2a).

A magnesium-chelated delafloxacin molecule is intercalated into the wedge-shaped space between the -1 and +1 nucleotides at each end of the 4-bp staggered DNA break site introduced into the 18-mer V-site duplex by nucleophilic attack of the two ParC Y118 residues (Supplementary Fig. 2). The 5′-phosphates at the ends of the DNA break are covalently linked to ParC Y118 residues confirming capture of the cleavage complex. Analysis of DNA conformation using the w3DNA programme revealed that all regions of the drug-captured G-gate DNA adopt a B-form helix (Supplementary Fig. 2a, b)[36]. Similar features and architecture were seen for complexes with the E-site DNA (Supplementary Fig. 2 and Supplementary Movie 1).

### Delafloxacin binds in a tilted ring conformation

In our topo IV-DNA structures, the $Mg^{2+}$ chelated delafloxacin molecules are bound in a highly unusual conformation in which strikingly the 6-amino-3,5-difluoro-2-pyridinyl substituent at N-1 is tilted 51.2° out-of-plane of the aromatic bicyclic quinolone core (Fig. 5a, b). The bulky C-8 chlorine is also out-of-plane by 6.8° and interacts with the face of the adjacent tilted 2-pyridinyl ring at N-1 (Fig. 5c). We were interested to know whether this unusual drug conformation is induced by enzyme binding or is intrinsic to the drug. No X-ray crystal structure of free delafloxacin is available, with previous work on related derivatives reporting difficulties in obtaining crystals suitable for X-ray analysis[37]. An X-ray crystal structure has been reported for a related compound (bearing a different N-1 aromatic ring and an N-ethylaminoazetidinyl C-7 substituent) revealing an N-1 ring tilted by 69° but was obtained for the p-TsOH salt, which complicates the analysis[37]. In the absence of a X-ray crystal structure of delafloxacin itself, we used computational chemistry methods (PM7 and Gaussian 16, see "Methods" section) to calculate the structure of the drug[38–40]. Very distinctive features of the computed structure are a tilted ring conformation with a rotation of the C-1 heteroaromatic group by 50.7° and an out-of-plane deviation of the C-8 chlorine bond of 17.1° (Supplementary Table 2). The near identical superposition of bound- and free-drug structures (Supplementary Fig. 3) suggests that topo IV binds the free drug conformation which entropic considerations suggest would favour tight binding[41].

### A 'water-metal ion bridge' stabilises drug binding

Our high-resolution structures reveal the complex anatomy and diverse interactions involved in drug binding. First, the most obvious feature is that the core quinolone ring system of each $Mg^{2+}$-chelated delafloxacin molecule is hemi-intercalated at each site of transient DNA cleavage between -1 and +1 nucleotides making π-π interactions with the +1-purine base (Figs. 4, 6–8 and Supplementary Figs 2 and 4–7). Although difficult to quantitate its importance for binding, drug intercalation obviously plays a crucial role in stabilising the cleavage complex by blocking rejoining of DNA ends.

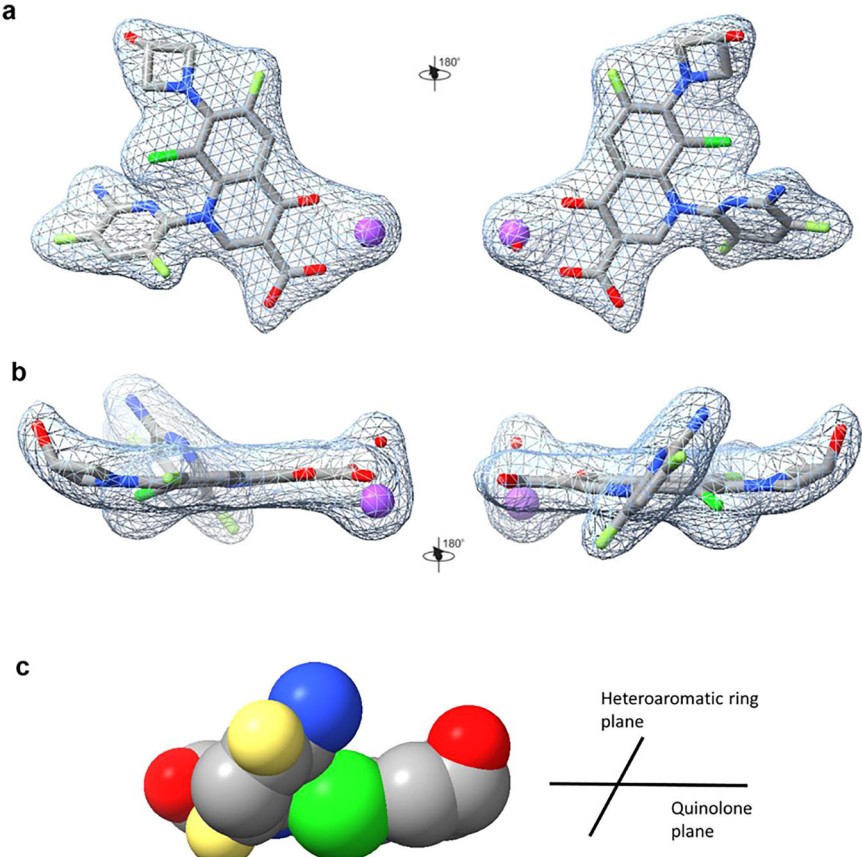

**Fig. 5 | Unusual conformation of delafloxacin in the 2.0 Å complex of *S. pneumoniae* topo IV with V18 DNA (PDB ID: 8QMB) featuring out-of-plane N-1 aromatic and C-8 chlorine substituents. a, b** Opposing views of the delafloxacin molecule with its chelated magnesium ion and associated waters with the electron density from the (Fo–Fc) map denoted by a mesh contoured at 1.5σ (limited to 2.3 Å range). The chelated magnesium ion is shown in purple and bound water shown in red. **a, b** show views perpendicular and parallel to the bicyclic quinolone ring, respectively. **c** Representation in full van der Waal radius of the delafloxacin molecule from the 2.0 Å topo IV X-ray crystal structure, indicating that the large electronegative Cl atom (shown in green) interacts directly to facilitate/stabilise tilting of the N1 heteroaromatic ring (dark grey).

Second, the drug chelated $Mg^{2+}$ ion engages in long range linkage (4.3-5.6 Å) to ParC and DNA bases via intervening water molecules (Fig. 7, and Supplementary Fig. 5). These linking waters are not visualised in lower resolution structures and are incompletely resolved in the 2.0 Å V-site complex (Fig. 7a). However, the near complete octahedral coordination sphere of the drug-bound $Mg^{2+}$ ion and associated links to enzyme and DNA are well-defined in the 2.4 Å E-site structure which has a different gate-DNA sequence (Fig. 7b, and Supplementary Fig. 5). Alongside the two chelating oxygens provided by the keto-carboxyl moiety of the drug, discrete electron density is seen for the oxygens of three metal ion coordinating waters. One of these water molecules clearly links the drug-bound $Mg^{2+}$ via hydrogen-bonding to the sidechain of ParC D83 (Fig. 7b, c) providing unequivocal structural evidence for the 'water-metal ion bridge' proposed to stabilise quinolone binding to conserved serine and/or acidic residues in ParC[22] and supported by mutagenesis and biochemical studies[42] (see also ref 2). The other two visible coordinating waters link the drug bound $Mg^{2+}$ to the +1-guanine and -1-thymine bases of the cleaved DNA (Fig. 7b, c and Supplementary Fig. 5). The sixth and final occupant in the $Mg^{2+}$ coordination shell is not imaged, but it could be a water molecule bridging to the hydroxyl group of ParC S79, a sidechain that also forms a direct hydrogen-bonded interaction with the drug C-3 carboxyl group as does the sidechain of ParC R117, the catalytic arginine residue (Fig. 7b, c). As discussed later, these features are important to a general understanding of fluoroquinolone action and target-mediated resistance arising from mutations at ParC S79 and D83.

## C-7 and N-1 groups anchor delafloxacin to topo IV

The delafloxacin substituents at C-7 and N-1 play an important role in binding the drug to ParE and ParC (Fig. 7a, b, Fig. 8). Although smaller than C-7 groups in other fluoroquinolones, the C-7 hydroxyazetidinyl moiety of delafloxacin maintains direct hydrogen bonds with the sidechain of ParE R456 and with the backbone peptide bond of ParE L412, interactions that are subtly different in the E-site complex (Fig. 7a, b). Moreover, the large tilted heteroaromatic group at N-1 is accommodated in a sub-pocket between the ParC subunit and DNA, forming a hydrogen bond with ParE D435 and a fluorine-oxygen bond with the phosphotyrosyl linkage involving the catalytic tyrosine, ParC Y118 (Figs. 7 and 8, Supplementary Figs. 6, 7 Supplementary Movie 2).

## Active site $Mg^{2+}$ ions and waters facilitate DNA cleavage

Reversible DNA breakage-reunion is a key activity of Top2As requiring divalent metal ions, usually assumed to be $Mg^{2+}$, that are distinct from the drug-bound $Mg^{2+}$ ions. Binding of a $Mg^{2+}$ to each scissile phosphodiester group on DNA (site A) is thought to promote DNA cleavage by each Y118 sidechain, with metal binding to the non-scissile phosphodiester between nts -1 and -2 (site B) stabilising the cleaved DNA. It is not clear whether both sites are simultaneously occupied or whether a single dynamic $Mg^{2+}$ moves between the sites[21,43-45]. We see that the delafloxacin-cleavage complex has two active site $Mg^{2+}$ ions- one per complementary DNA strand and each located at the B site (Fig. 8). These $Mg^{2+}$ ions lie above the drug binding pockets and are chelated by a triad of acidic residues from the ParE TOPRIM domain, namely ParE

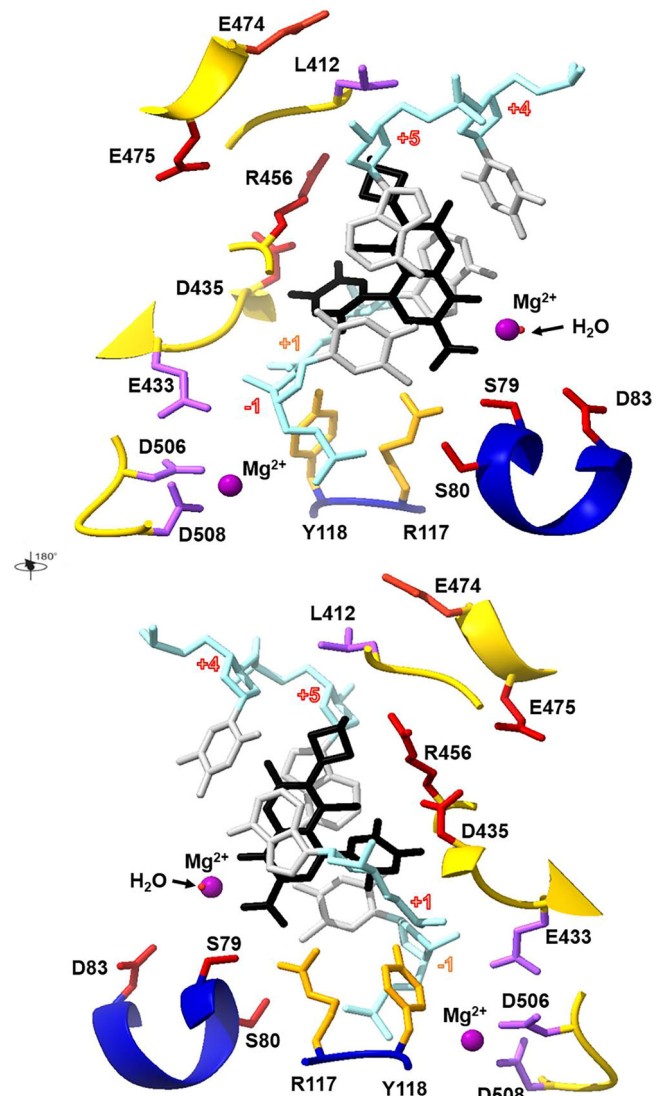

**Fig. 6 | Details of the topo IV-DNA binding site for delafloxacin, one of two sites present in the 2.0 Å X-ray crystal structure with the V18 gate-DNA (PDB ID: 8QMB).** The figure shows opposing views of the drug binding site. The ParC backbone is in blue, the ParE backbone is in yellow, the DNA backbone is in light blue, the drug molecule is shown in black, and magnesium ions are shown in purple. Active site tyrosine (Y118) and arginine (R117) sidechains from ParC are in orange, DNA bases and sugars are in silver. The active site magnesium-coordinating residues are in purple, and the ParC S79 and D83 residues whose mutation is associated with drug resistance are shown in red.

D506 and D508 (from the IMTDxD motif) and E433 (Fig. 8). Intercalation of the fluoroquinolone pushes the reactive 5′-phosphotyrosyl and 3′-hydroxyl ends far apart, ties up the catalytic ParC R117 and stabilises/displaces the active site $Mg^{2+}$ ions so that they coordinate the non-scissile phosphodiester group between nucleotides -2 and -1 (Fig. 9). The structures of all fluoroquinolone cleavage complexes determined to date by X-ray crystallography or by cryo-electron microscopy have the two B-sites occupied by $Mg^{2+}$ ions whereas in drug-free pre-cleavage complexes the two $Mg^{2+}$ ions occupy the A sites[21,43–46].

Interest in metal ion catalysis by topoisomerases has focused attention on the positioning and detailed coordination of the A and B site metal ions in the various functional states of Top2A enzymes. In this regard, the 2.0 Å delafloxacin complex is highly informative as all six positions in the coordination sphere of the active site $Mg^{2+}$ ions are clearly resolved (Fig. 9). Two of the six inner coordination sites are

occupied directly by sidechain carboxyl oxygens from ParE D508 and D506. A third carboxyl oxygen from ParE D433 is linked to the $Mg^{2+}$ through a clearly visible inner sphere water. The remaining three inner coordination positions are occupied by three water molecules, one of which links to the non-scissile phosphodiester group joining nucleotides at -2 and -1, with a second water potentially bridging to the more distant phosphotyrosyl formed by the catalytic ParC Y118. In the presence of bound drug, the $Mg^{2+}$ ion coordinates the non-catalytic phosphodiester group linking -1 and -2 nucleotides and is therefore unable to participate in DNA resealing but consistent with a role in stabilising the cleaved DNA state. Clearly, the new high-resolution structures provide a wealth of detail about $Mg^{2+}$-DNA coordination in the context of the drug-arrested cleavage complex.

### New-found potassium ions in the cleavage complex

By measuring the anomalous signals observed in tunable long wavelength X-ray diffraction of a relatively low resolution (2.62 Å) V-site-delafloxacin core complex (PDB 9GEF), we could verify the assignment of the two drug-bound- and two active site-metal ions as $Mg^{2+}$ in our the earlier 2.0 and 2.4 Å structures described here (PDB 8QMB and 8QMC)[47]. However, scrutiny of the 2.0 Å and 2.4 Å structures and our previous deposited structures of pneumococcal gyrase and topo IV (PDB 4I3H, 4KOE,4Z4Q) also revealed the presence of two additional metal binding sites located one to each 'tower' domain and each formed by a ParC FKYTDLQIN motif (Figs. 4, 10). Our subsequent studies established that these sites are occupied by potassium ions evidenced by the difference in the anomalous potassium signal below and above the potassium K-edge and which had been misassigned as water or $Mg^{2+}$ in these and other structures (including 5BS8, 4Z53 and 5CDQ among others)[47]. Each potassium ion interacts with the main chain or sidechain of a ParC subunit through F316, K317, T319, E320, Q322 and N324 and involves a complex network of coordinating water molecules that are nicely imaged (Fig. 10). The $K^+$ ions stabilise a ParC loop in which the metal ion caps an α-helix lying near the surface of the 'tower' adjoining and potentially stabilising the bound G-gate DNA in the DNA binding groove (Figs. 4, 10). It is known that gyrase requires $K+$ ions for supercoiling activity[48] with recent X-ray crystal structures of the *E. coli* gyrase ATPase domain revealing a $K^+$, $Na^+$ and $Mg^{2+}$ ion associated with each enzyme bound ATP molecule[49]. Our core cleavage complexes of topo IV lack the ATPase domain and therefore the $K^+$ ions that we see must act differently. These results provide clear evidence of potassium ions in the cleavage core of a type II topoisomerase.

## Discussion

Delafloxacin is a structurally unique fluoroquinolone recently approved for the treatment of pneumococcal pneumonia. An intriguing feature of the drug is its clinical potency including activity against bacterial strains resistant to other therapeutically used fluoroquinolones. Given the medical interest in combatting fluoroquinolone resistance, we have sought to understand how delafloxacin binds its topoisomerase targets *vis-á-vis* other fluoroquinolones. Our high-resolution structures of delafloxacin-DNA cleavage complexes of *S. pneumoniae* topo IV solved here at 2.0 Å and 2.4 Å (i.e., with V- and E-site DNA and assisted in part by using a novel graphene nucleant) can be used to address this issue (Fig. 4, and Supplementary Fig. 2 and Supplementary Table 1). Analysis has provided important drug-specific and generic insights on fluoroquinolone binding that were not evident from earlier low-resolution structures involving other fluoroquinolones. Key findings include the unexpected out-of-plane conformation of two interacting substituents on the bound delafloxacin molecules, unique delafloxacin-specific enzyme contacts, visualisation of the full coordination spheres of drug-bound- and active site-magnesium ions, the direct imaging of multiple long-range water-$Mg^{2+}$ ion links that coordinate drug, protein and DNA, and

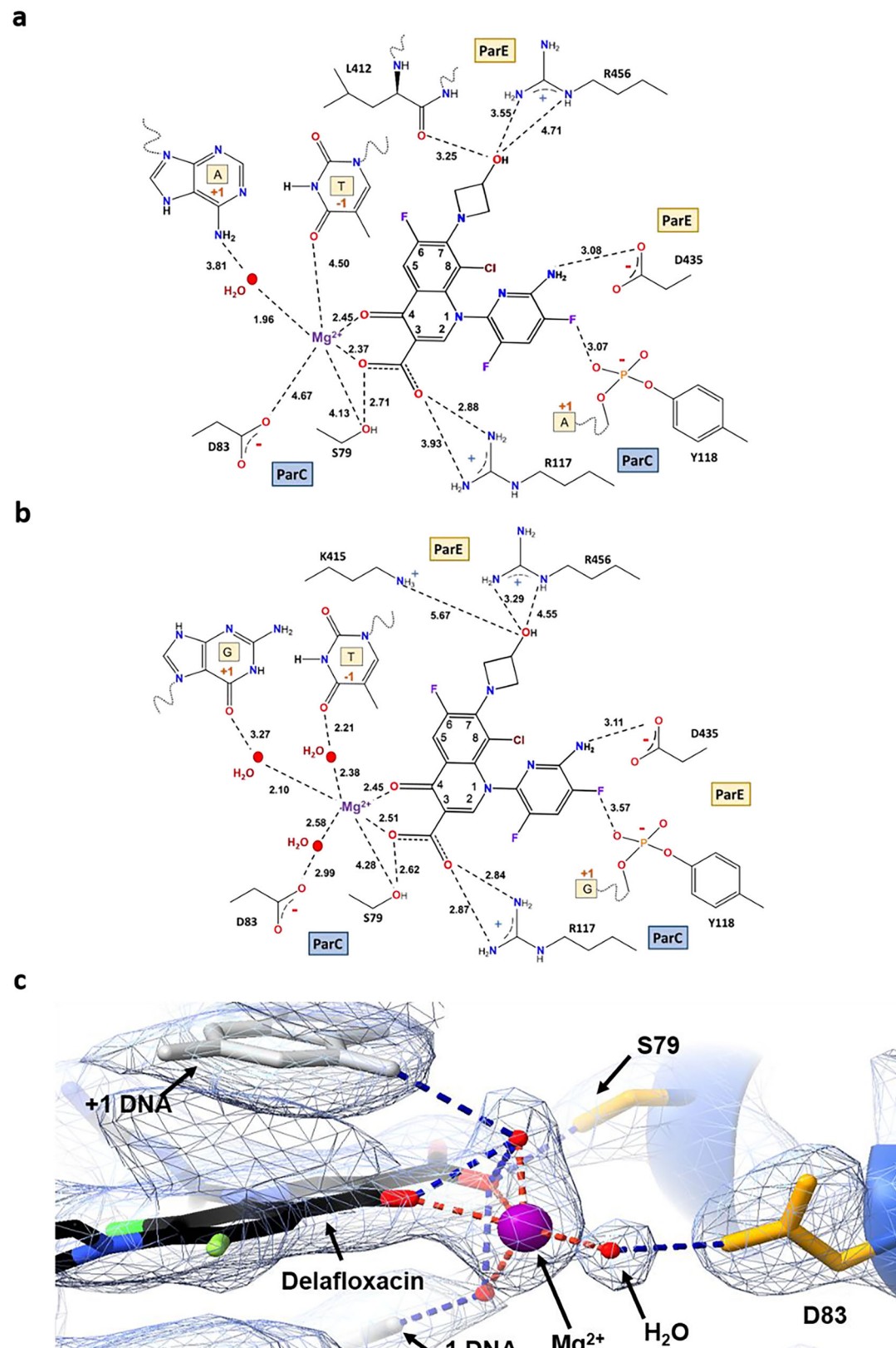

**Fig. 7 | A magnesium ion coordinates drug binding to topo IV and DNA via multiple water-metal ion links. a, b** Schematic representation of the water coordination spheres of delafloxacin chelated $Mg^{2+}$ derived from 2.0 Å V-18 (**a**) and (better resolved) in 2.4 Å E18 (**b**) X-ray crystal structures of DNA cleavage complexes with topo IV (PDB ID: 8QMB). Important distances and likely hydrogen bonds are shown by dashed lines; only water molecules that could be individually imaged are shown (in red). **c** Imaging of the water molecule that forms a water-metal ion bridge between the delafloxacin and the ParC D83 sidechain (2mFo-DFc map contoured at 1.5σ) (PDB ID: 8C41).

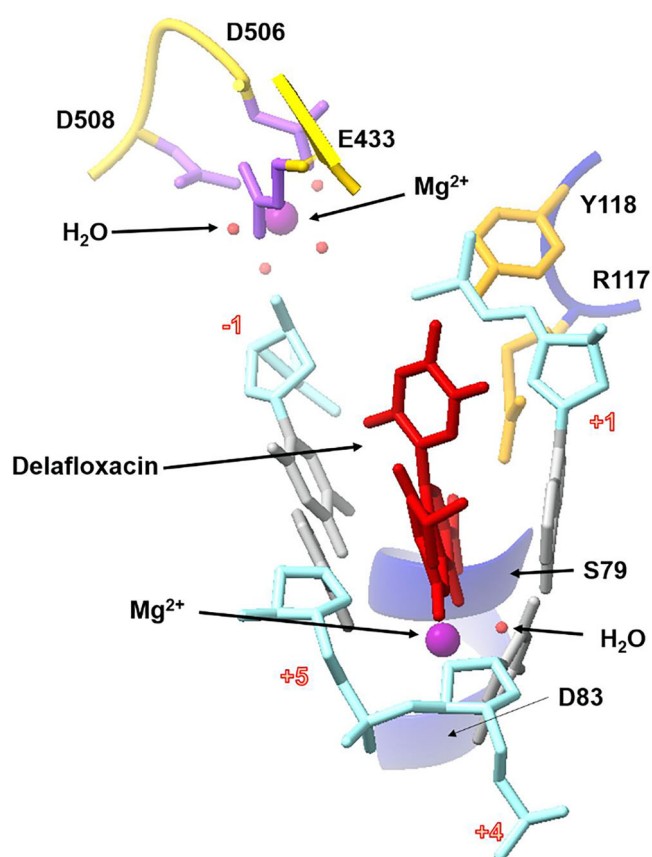

**Fig. 8 | Top-down view of the DNA cleft occupied by delafloxacin in the 2.0 Å V−18 X-ray crystal structure (PDB ID: 8QMB).** The ParC backbone is in blue, the ParE backbone is in yellow, DNA backbone is in light blue, the intercalated drug molecule is in red, magnesium ions are in purple, and the active site tyrosine (Y118) and arginine residues (R117) are in orange, DNA bases and sugars are in silver, active site magnesium-coordinating residues are in purple. The positions of ParC S79 and D83 residues responsible for quinolone resistance upon mutation are indicated by arrows.

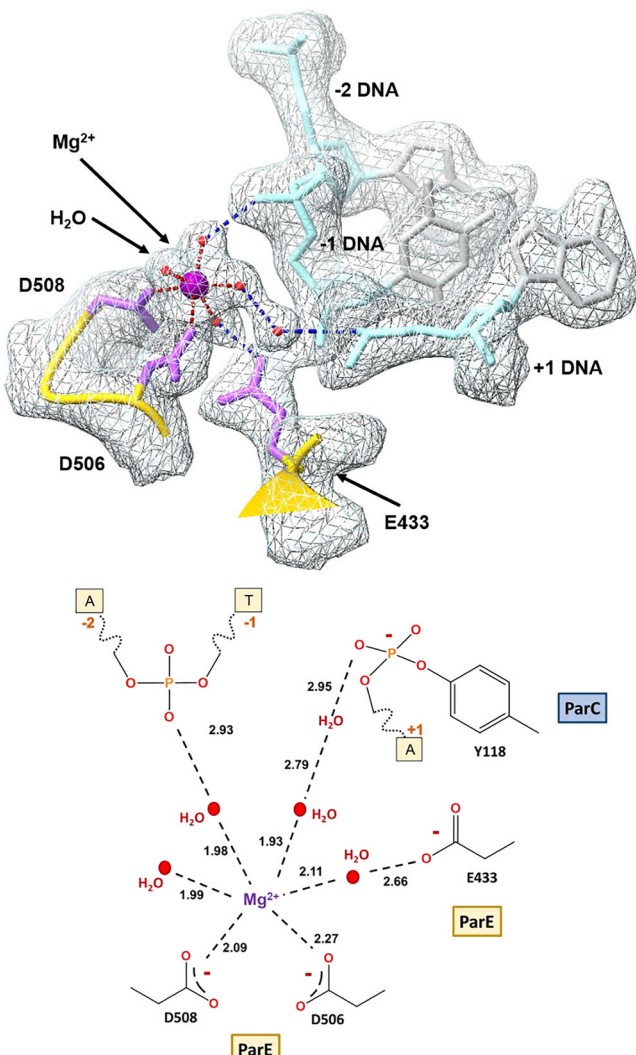

**Fig. 9 | Electron density and schematic diagram showing the full coordination sphere and multiple water mediated linkages of the active site magnesium in the 2.0 Å delafloxacin-core complex X-ray crystal structure (PDB ID:8QMB),** with the 2Fo−Fc electron density map contoured at the 1.5σ level. ParE back-bone is in yellow, DNA backbone is in light blue, magnesium ions are in purple, ParE residue side chains are in light purple, DNA bases and sugars are in silver, coordinated water molecules in red. Metal coordination is shown by dashed red lines; likely hydrogen bonds between the coordinated water molecules and the residues are shown as dashed dark blue lines. Bottom figure: Mg2+ coordination and links directly or via water molecules to protein or DNA are shown by black dashed lines.

the unexpected discovery of potassium ions that may aid assembly of the cleavage complex. Together with biochemical and computational studies, our work on delafloxacin provides the best understanding to date of how this fluoroquinolone binds its enzyme-DNA target and helps explain its differential antibacterial potency (Fig. 3)[4,8,9].

As with other fluoroquinolones, two delafloxacin molecules intercalate between -1 and +1 nucleotides at the two ends of a transient DNA break, thereby blocking DNA resealing and stabilising the topo IV/gyrase cleavage complex (Figs. 4 and 8, Supplementary Fig. 2). However, compared to other therapeutic congeners, the conformation of the topo IV-bound delafloxacin is highly unusual with the N-1 aromatic ring tilted by 51.2° and with the C-8 chlorine also bent slightly out-of-plane (Figs. 5, 6, and Supplementary Fig. 7 and Supplementary Table 2). Essentially the same non-sp² conformation was obtained for the free drug using a sophisticated computational analysis (Supplementary Fig. 3, and Supplementary Table 2). We suggest that the direct interaction of the bulky Cl with the face of the nearby N-1 heteroaromatic ring (perhaps aided by its two electron withdrawing fluorine substituents) may stabilise this tilted ring conformation or sterically block its rotation (Fig. 5c). It is known that fixing the conformational state of a drug favours tight target binding by minimising the entropic cost entailed in selective binding of a preferred conformation such that the full ligand binding energy can be utilised[41]. So-called 'conformational constraint' of a flexible ligand is a commonly used strategy in drug development that promotes tighter binding to enhance potency and

target selectivity[50]. Currently, the factors responsible for the greater quinolone sensitivity of topo IV over gyrase are unknown and must await the solving of a high-resolution structure of a pneumococcal gyrase cleavage complex with delafloxacin to facilitate comparison with that of topo IV.

The unusual conformation of the topoisomerase-bound delafloxacin led us to revisit our earlier deposited E-site structures of pneumococcal topo IV and gyrase cleavage complexes formed with trovafloxacin, a potent investigational fluoroquinolone comprising a naphthyridone ring bearing a related aromatic (2-4-difluorophenyl) group at N-1 (Fig. 1). These structures (e.g., PDB 4KOE, 4Z2E) reveal that the N-1 aromatic group of trovafloxacin bound to topo IV or gyrase is also out of plane by a similar angle to that seen with delafloxacin (see overlay in Supplementary Fig. 8). Moreover, this tilted conformation is present in undeposited X-ray crystal structures of free trovafloxacin (and of the related drug tosufloxacin) (Supplementary Fig. 1)[51]. We

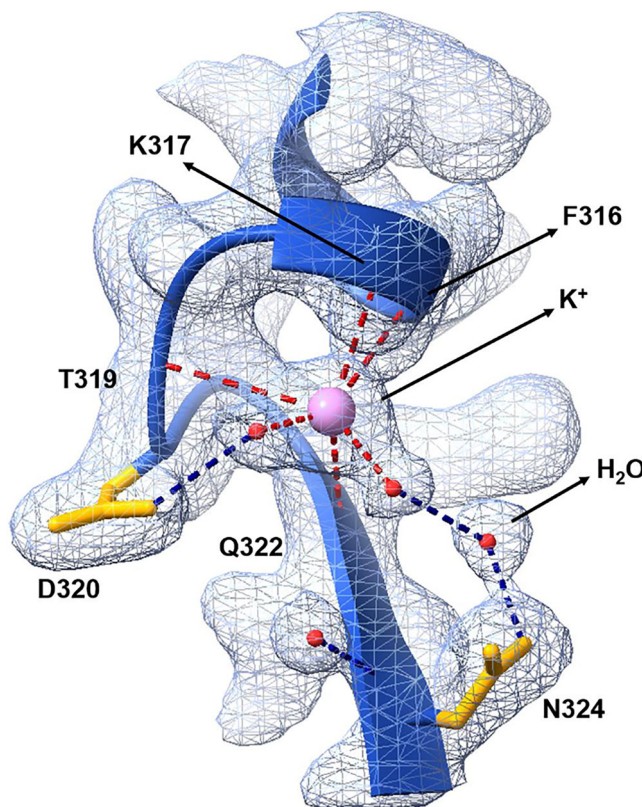

**Fig. 10 | Coordination sphere and binding motif of the newly found surface potassium ions located one to each ParC tower domain.** The figure is from the 2.0 Å V18 delafloxacin core complex X-ray crystal structure (PDB ID: 8QMB) with the 2Fo−Fc electron density map contoured at the 1.5σ level. The ParC backbone is in blue, ParC sidechains are in orange, the potassium ion is in pink and coordinated water molecules are in red. Metal coordination is shown by dashed red lines, and likely hydrogen bonds between the coordinated water molecules and protein residues are denoted by dashed dark blue lines.

speculate that, as with the C-8 Cl of delafloxacin, the lone pair of electrons on the N-8 nitrogen of trovafloxacin may interact with the N-1 aromatic ring to fix its tilted conformation and similarly enhance enzyme binding.

We were intrigued that delafloxacin and other fluoroquinolones with aromatic N-1 substituents adopt an out-of-plane conformation rather than the $sp^2$ planar geometry predicted of these multi-aromatic ring systems. However, a search of the Cambridge Crystallographic Data Centre of small molecules using a generic framework comprised of a quinolone nucleus and aromatic N-1 substituent identified 178 entries producing the distribution of torsion angles seen in Supplementary Fig. 9. About half of the entries show significant tilting of the N-1 ring (non-zero torsion angles) indicating that steric and other factors can seemingly override electronic contributions to generate non-planar geometry.

Delafloxacin has a core quinolone ring system with two strongly electronegative substituents: a fluorine at C-6 and a chlorine at C-8 (Fig. 1). Historically, incorporation of fluorine was crucial to the development of the modern antimicrobial quinolones with superior activity[1,2]. The addition of the C-8 Cl further improves antibacterial potency and spectrum[1]. It is believed that the strongly electron-withdrawing fluorine (and by analogy the C-8 Cl) polarises the drug molecule and promotes drug intercalation by modulating π-π inter-actions between the aromatic bicyclic quinolone nucleus and the DNA bases. Thus, the Cl substituent in delafloxacin has potentially dual effects i.e., on the drug dipole and on the drug conformation. The dipole effect alone may operate in simpler chlorofluoroquinolones

lacking an aromatic N-1 group such as besifloxacin, used in treating pneumococcal conjunctivitis, and in the potent experimental agent clinafloxacin (Fig. 1, and Supplementary Fig. 1) used to solve the structure of a fluoroquinolone-arrested cleavage complex (PDB 3FOE)[20,52,53].

One interesting question is how a variety of structurally diverse fluoroquinolones and other unrelated drugs such as zoliflodacin and gepotidacin can stabilise the Top2A DNA cleavage complex. This issue is relevant for delafloxacin with its bulky N-1 group and small ring at C-7, i.e., the steric inverse of other therapeutic fluoroquinolones such as moxifloxacin and levofloxacin (Fig. 1). It seems that the fluor-oquinolone binding pocket can accommodate a range of substituents at N-1 and C-7 allowing drug intercalation at the cleavage site (Figs. 6, 8, and Supplementary Fig. 6). For delafloxacin, the small hydroxy-azetidinyl group still forms a productive hydrogen bonded contact with ParE. Moreover, an unsuspected side pocket accommodates the bulky N-1 aromatic group whose increased surface area and additional enzyme contacts can augment target affinity in a way denied quino-lones such as levofloxacin and moxifloxacin that have small inert N-1 substituents (Figs. 7, 8). Interestingly, quinazolinediones and spir-opyrimidinetriones like zoliflodacin use the quinolone binding pocket but have their principal enzyme contacts with ParE/GyrB so they are immune to the effects of quinolone resistance mutations in ParC/GyrA[21,25,26,28]. A different means of bypassing quinolone resistance is achieved by allosteric inhibitors that stabilise the cleaved DNA gate by binding at remote sites and by NBTIs such as gepotidacin that trap the DNA gate by binding at a different enzyme-DNA interface[30,31]. In accommodating these varied inhibitors, it appears that topo IV and gyrase exhibit a degree of conformational flexibility, in keeping with their function as macromolecular machines (Fig. 2).

Our high-resolution structures reveal the crucial role that metal ions and water molecules play in Top2A catalysis and drug inhibition. Early structural work indicated that fluoroquinolone-arrested cleavage complexes of topo IV and gyrase contain a pair of 'drug-bound' $Mg^{2+}$ ions and a pair of active-site $Mg^{2+}$ ions that together coordinate a complex network of short- and longer-range interactions with enzyme and DNA[20–24]. These links have remained incompletely defined due to the generally poor resolution (≥3 Å) of the structures available to date. However, our 2.0 Å V-site- and 2.4 Å E-site structures of the delafloxacin-topo IV cleavage complex are at a resolution wherein more than 1000 water molecules are individually imaged. Conse-quently, we were able to visualise the long range links formed by the drug-bound $Mg^{2+}$ via intervening water molecules to the +1 and -1 DNA bases at the cleavage site (possibly contributing to the DNA cleavage specificity of topo IV) and to ParC D83 (Fig. 7, Supplementary Fig. 7)[19,34]. Furthermore, the active site $Mg^{2+}$ ions (present at the B site on each DNA strand) interact directly with ParE D506 and D508 but we could also image their longer range interactions via intervening water molecules to ParE E433, to the 5′ phosphotyrosyl group, and to the non-scissile phosphodiester group linking -1 and -2 nucleotides, thereby stabilising the post-cleavage state (Fig. 9). Thus, our structures reveal the key roles played by water molecules in coordinating activ-ities across the topo IV complex.

An unexpected development arising from our initial character-isation of magnesium binding sites in this work was the discovery and confirmation by anomalous scattering[47] that the topo IV cleavage complex contains a pair of symmetrically disposed $K^+$ ions (Figs. 2, 4, and 10). These $K^+$ ions, misassigned as $Mg^{2+}$ in earlier structural depositions to the PDB, are bound one to each 'tower' domain of pneumococcal topo IV through a specific ParC sequence (L)FKYTDL-QIN that forms an α-helix-loop-beta sheet motif located close to the G-gate DNA. This sequence and its three-dimensional fold are highly conserved across other bacterial topo IV and gyrase complexes (Fig. 4, Supplementary Fig. 10a, b). $K^+$ ions are positively charged and could bind and perhaps help bend the gate-DNA through interaction with

DNA phosphates. Our structures of pneumococcal topo IV cleavage complexes determined here used an 18-mer gate DNA which is too short to bridge to the K⁺ ion. However, overlay of structures determined for DNA cleavage complexes formed with different topo IV and gyrases and a longer (≥21 bp) gate DNA indicate that the K⁺ is close enough (~3 Å) to bind a phosphate oxygen between nucleotides 20 and 21 thereby stabilising the bound DNA (Supplementary Fig. 10b).

Superimposition of available Top2A structures on that of the pneumococcal topo IV-DNA cleavage complex and analysis by RMSD indicated that bacterial and eukaryotic Top2A enzymes are closely similar in overall 3-D structure (Supplementary Table 3). The tower domain protein sequence equivalent to the potassium binding region is highly conserved among eukaryotic enzymes but has only limited homology to its bacterial counterparts (Supplementary Fig. 10a). Nonetheless, in all cases, the eukaryotic topo II tower sequence adopts an α-helix-loop-beta sheet motif that maps on the K⁺ binding site in pneumococcal topo IV, though with a less pronounced loop region and greater distances between the K⁺ ion and a DNA phosphate oxygen (Supplementary Fig. 10c). To our knowledge, there is no published X-ray crystal structure of a eukaryotic Top2A carrying a metal ion at this site perhaps reflecting a lack of potassium salt inclusion in many crystallisation mixes. Clearly more work will be needed to ascertain whether other Top2As contain K⁺ and to establish functional and/or structural roles e.g., by site-directed mutagenesis. We note that a second pair of K⁺ ions bridging ParE and ParC subunits revealed by our separate anomalous scattering studies was not imaged in the present structures[47]. The upshot of our work is that topo IV joins the ribosome as another molecular machine in which K⁺ ions are structurally associated with a catalytic centre[54].

Water-metal ion bridges are of particular interest in relation to the growing problem of target-based clinical resistance to fluoroquinolones[1,55]. Resistance commonly arises through mutation of two highly conserved Ser and Asp/Glu residues present in both ParC and GyrA. Mutations at these loci are thought to disrupt the water-metal ion bridge linking the fluoroquinolone to one or both ParC/GyrA residues thereby reducing drug affinity and requiring higher drug concentrations to trap the cleavage complex[2,42]. However, as described in detail in ref 2, depending on the enzyme and bacterial species, the bridge may instead play a drug positioning rather than binding role as is seen for both E. coli gyrase and topo IV. Evidence implicating water-metal ion bridges came initially from analysis of a 3.27 Å resolution structure of a moxifloxacin-stabilised E-site cleavage complex of Acetinobacter baumannii topo IV[22]. This work indicated the presence a water-Mg ion bridge involving both ParC S84 and E88 residues (corresponding to ParC S79 and GyrA D83 in pneumococcal topo IV). However, at that resolution, the water molecules could not be directly imaged but rather their positions in the electron density were interpolated from a difference map (with and without drug) using the known octahedral coordination geometry of the drug-bound Mg²⁺ ion. Subsequently, from a 2.4 Å E-site structure of a moxifloxacin cleavage complex of the naturally resistant Mycobacterium tuberculosis DNA gyrase, a bridging water molecule was visualised bound to the relevant serine residue which had been reverse engineered to replace the natural resistance-conferring GyrA alanine at position 90 (equivalent to S. pneumoniae ParC S79)[33]. Complementing these gyrase studies, our 2.4 Å delafloxacin E-site structure now provides direct and convincing visual evidence for topo IV of a discretely imaged water-metal-ion bridge to ParC D83 (Fig. 7c, and Supplementary Fig. 5). It is known that depending on the bacterial species, enzyme and fluoroquinolone, water-metal ion bridges can involve drug linkage to either or both TOPRIM residues[2,42].

It may seem surprising that mutational loss of a single hydrogen bond to serine or aspartate in topo IV/gyrase can lead to clinical resistance, given that hydrogen bonds are weak in an aqueous environment. However, work on vancomycin-resistant strains has shown that loss of a single hydrogen bond by modification of its peptidoglycan substrate reduces both drug binding and vancomycin activity by 10³-fold[56]. For pneumococcal topo IV and gyrase reconstituted with quinolone-resistant ParC S79F and GyrA S81F subunits, cleavable complex formation was typically reduced by 8-16-fold for all fluoroquinolones tested compared with the wild-type enzyme[57]. Cleavage was restored at higher drug levels, indicating that the mutations do act by reducing drug binding. Presumably, loss of the water-metal ion bridge to ParC/GyrA pivotal for clinical activity of less potent fluoroquinolones, is less important for drugs such as delafloxacin that have other physiological and molecular advantages.

Delafloxacin is one of a small number of anti-pneumococcal quinolones that have made it into clinical use. It is also one of the most potent. Previous studies have shown that delafloxacin was 64- to 128-fold more active against S. pneumoniae isolates than levofloxacin[58,59]. When tested against invasive S. pneumoniae isolates including highly levofloxacin resistant strains, it was 64 times more potent than levofloxacin[8]. These differential effects may at first seem paradoxical given that fluoroquinolones all share the same core structure comprising a bicyclic quinolone (or naphthyridone) ring system, a 3,4-carboxyketone moiety (needed to bind a Mg²⁺ ion), a fluorine substituent at C-6, and absent or small substituents at C-2 and C-5 (Fig. 1). Adherence to this structural core ensures intercalation of the Mg²⁺-chelated drug into DNA and further stabilisation via water-mediated links to DNA and to ParC/GyrA. However, it is the presence of various drug-specific groups at N-1, C-7 and C-8 that results in differential activity among fluoroquinolones through their potentially different effects on plasma protein binding, half-life, tissue penetration, bacterial uptake/efflux, target affinity, and gyrase versus topo IV targeting. Delafloxacin has particularly interesting N-1 pyridinyl, C-7 azetidinyl and C-8 Cl groups that we show by structural, biochemical and computational means can potentiate binding to the topo IV cleavage complex, in ways not available to other fluoroquinolones. These bespoke interactions account for the much greater potency of delafloxacin over levofloxacin in trapping the topo IV cleavage complex in vitro (Fig. 3), a property also seen in its 60-fold lower MIC against wild-type and levofloxacin-resistant pneumococcal isolates bearing mutations in both parC and gyrA genes (FDA Baxdela NDA208610 and 208611, 2019). The weakening of a single water-metal ion bridge through mutation at S83 or D87 in ParC or GyrA reduces drug binding and at most leads to an 8-16-fold decrease in topo IV/gyrase cleavage in vitro with the bacteriological effects of single mutations reduced for dual targeting drugs (that act equally on both enzymes as observed for delafloxacin in E. coli and S. aureus)[7]. In fact, many resistant clinical isolates have a mutation in both targets, increasing the MIC for many quinolones by 32- to 64-fold. For less potent fluoroquinolones, mutation of one or both targets can compromise medical efficacy. However, enhanced target affinity and/or altered pharmacological properties (e.g., enhanced uptake under acidic conditions) of potent quinolones such as delafloxacin, clinafloxacin and gemifloxacin can potentially offset these effects to ensure a bactericidal outcome whereby even the double mutants remain within the clinically effective concentration range of the drugs[53]. These considerations rationalise clinical findings and indicate that alongside its anionic character favouring bacterial uptake, enhanced target affinity likely plays an important role in delafloxacin action.

In this paper, we present the structures of a bacterial Top2A DNA cleavage complex formed with the anionic fluoroquinolone delafloxacin and solved at 2.0 and 2.4 Å, the highest resolution achieved to date for any fluoroquinolone complex. We show that the two intercalated drug molecules have an unusual conformation involving interacting out-of-plane aromatic and chlorine substituents and participate in unique interactions with the topo IV enzyme and DNA. We identify the enhanced intrinsic potency of delafloxacin-demonstrated here in DNA cleavage assays, coupled perhaps with acid-promoted

bacterial uptake, as a key factor that can explain therapeutic activity against target-mutated bacteria resistant to other fluoroquinolones. Moreover, the higher resolution topo IV structures have revealed the full coordination shells of drug-bound- and active site- magnesium ions as well as newly uncovered potassium ions and have allowed imaging of water-mediated hydrogen bonds relevant to topoisomerase catalysis and fluoroquinolone action. Our structural, biochemical, and computational work enriches understanding of delafloxacin and fluoroquinolone action. Growing appreciation of the rich structural and functional diversity of topo IV and gyrase in different bacterial pathogens and the important role of the disease microenvironment may aid the design of new fluoroquinolones effective against drug-resistant disease.

## Methods

### plasmids and DNA substrates

Expression plasmids pXP9, pXP10, pXP13, pXP14 and pET29aParC55 were made in our laboratory as were plasmids pXP1 and pET29aParC55[60,61]. The plasmid pET29aParE30ParC55 encoding the fusion protein was constructed using pXP1 (which carries a 4.3-kb HindIII fragment of the *S. pneumoniae parE-parC* locus) and pET29a-ParC55 (which expresses *S. pneumoniae* ParC55 as a C-terminally His$_6$-tagged protein). These plasmids were from our laboratory collection. *Escherichia coli* XL-1 Blue supercompetent cells and *E. coli* BL21 (λDE5) pLysS were purchased from Stratagene and Novagen, respectively. Supercoiled plasmid pBR322 used for DNA cleavage assays was obtained from New England Biolabs. All oligonucleotides were synthesised by solid phase phosphoamidite chemistry and were doubly HPLC purified by Metabion, Munich.

### Chemicals and reagents

Delafloxacin and levofloxacin were obtained from Sigma-Aldrich and McNeil-Ortho, respectively. SYBR Green 1 Nucleic Acid Gel Stain was sourced from Lonza, Slough, UK. NTA-resin was from Qiagen. BioTAQ DNA polymerase was from Bioline and Pfu DNA polymerase was from Stratagene.

### Expression of topo IV and gyrase proteins

Full-length *S. pneumoniae* gyrase GyrB and GyrA and topo IV ParC and ParE subunits were separately expressed from plasmids pXP9, pXP10, pXP13 and pXP14[60]. Briefly, *E. coli* strain BL21(λDE3) pLysS was transformed with the appropriate expression plasmid, subunit expression was induced by IPTG and the recombinant gyrase and topo IV proteins were purified to >95% homogeneity by nickel chelate (NTA-resin) chromatography as in our published methods[60].

To obtain the ParE30-ParC55 fusion protein, we constructed an expression plasmid starting from pET29a-ParC55, a plasmid we engineered to enable the inducible expression of a C-terminally His6-tagged ParC55 protein. Importantly, this plasmid has a single NdeI site that overlaps the N-terminal (Met) initiation codon of the ParC55 gene allowing the insertion of the ParE30 coding sequence directly upstream of the *parC55* gene sequence generating an inducible gene that expresses the fusion protein. Specifically, the ParE30 DNA coding sequence was amplified by PCR using *parE*-containing plasmid pXP1 DNA as template, the proofreading Vent DNA polymerase, and two primers each containing an NdeI restriction sequence (underlined):

Forward primer 5′-GGTTC*CATATG*AAAAACAAGAAAGATAAGGG CTTG

Reverse primer 5′-AAAA*CATATG*AAACACTGTCGCTTCTTCTAG CGT

The resulting 730 bp PCR product was digested with NdeI and ligated into NdeI-linearised pET29a-ParC55 prior to transformation of competent *E. coli* cells. Plasmids were purified from transformants and checked by DNA sequence analysis to confirm the correct insert orientation and the absence of mutations in the coding

region. The overexpression plasmid was transformed into *E. coli* host BL21(λDE3) pLysS. The procedure for overexpression and purification of the C-terminal His tagged fusion protein to >95% homogeneity followed what we described previously for the N-terminally His- tagged-ParE30 protein[21]. The linkage strategy added an extra histidine residue between ParE30 and ParC55 to generate an 84 kDa fusion protein with 742 amino acid resides including 6 C-terminal histidines:

```
MKNKKDKGLLSGKLTPAQSKNPAKNELYLVEGDSAGGSAKQGRDRKFQAILPL
RGKVINT 60
AKAKMADILKNEEINTMIYTIGAGVGADFSIEDANYDKIIIMTDADTDGAHIQ
TLLLTFF 120
YRYMRPLVEAGHVYIALPPLYKMSKGKGKKEEVAYAWTDGELEELRKQFGKGA
TLQRYKG 180
LGEMNADQLWETTMNPETRTLIRVTIEDLARAERRVNVLMGDKVEPRRKWIED
NVKFTLE 240
EATVFHMSNIQNMSLEDIMGERFGRYSKYIIQDRALPDIRDGLKPVQRRILYS
MNKDSNT 300
FDKSYRKSAKSVGNIMGNFHPHGDSSIYDAMVRMSQNWKNREILVEMHGNNGS
MDGDPPA 360
AMRYTEARLSEIAGYLLQDIEKKTVPFAWNFDDTEKEPTVLPAAFPNLLVNGS
TGISAGY 420
ATDIPPHNLAEVIDAAVYMIDHPTAKIDKLMEFLPGPDFPTGAIIQGRDEIKK
AYETGKG 480
RVVVRSKTEIEKLKGGKEQIVITEIPYEINKANLVKKIDDVRVNNKVAGIAEV
RDESDRD 540
GLRIAIELKKDANTELVLNYLFKYTDLQINYNFNMVAIDNFTPRQVGIVPILS
SYIAHRR 600
EVILARSRFDKEKAEKRLHIVEGLIRVISILDEVIALIRASENKADAKENLKV
SYDFTEE 660
QAEAIVTLQLYRLTNTDVVVLQEEEAELREKIAMLAAIIGDERTMYNLMKKEL
REVKKKF 720
ATPRLSSLEDTAKALEHHHHHH 742
```

EGDSA and IMTDxD sequences in ParE are underlined and their Mg$^{2+}$-chelating residues are shown in bold; ParC S79 and D83 residues altered in quinolone resistance, and the catalytic tyrosine ParC Y118 are in bold and underlined. An FKYTDLQIN potassium binding motif in ParC reported here is underlined. An extra histidine (introduced in linking ParE30 and ParC55 coding sequences in the expression plasmid) is italicised and underlined.

### Preparation of DNA substrates

Complementary 18-mer oligomers for the E site (5′-dCATGAATGAC-TATGCACG-3′ and 5′-dCGTGCATAGTCATTCATG-3′) and for the V site (5′-dTGTGGATAACCGTATTAC-3′ and 5′-dGTAAACGGTTATCCACA-3′) were doubly purified by HPLC and used to prepare the respective G-gate DNAs for crystallographic work. The DNA duplexes were formed by dissolving the complementary oligomers in annealing buffer (10 mM Tris-HCl, pH 7.5, 200 mM NaCl, 5 mM 2-mercaptoethanol, 0.05% NaN$_3$) in equimolar amounts, heating to 95% and allowing to cool slowly to 4 °C over a period of 48 h in a Dewar (a sealed Thermos flask). This procedure was adopted to prevent the formation of DNA hairpins by the single-stranded DNA oligos and to drive the equilibrium toward duplex formation.

### Crystallisation

Fused-ParC55-ParE30 protein was concentrated to 4.5 mg/ml and dialysed into 20 mM Tris-HCl (pH 7.5), 200 mM NaCl and 0.05% NaN$_3$. The protein was then mixed with the 18-mer E site or V site duplex DNA at a 1:1.2 molar ratio. Delafloxacin (dissolved in diluted NaOH) and MgCl$_2$ were added to a final concentration of 2 mM and 10 mM, respectively. The mixture was incubated at room temperature overnight to allow complex formation.

A Mosquito robot (supplied by TTP Labtech) was used to set up crystallisation trials for the cleavage complexes. Trials were performed in MRC 96-well 2-Drop plates using the sitting-drop vapour diffusion technique. The drops consisted of a mixture of 600 nl of the protein complex and 400 nl of the precipitant solution and were equilibrated against a 50 μl reservoir. In the case of V-site 18-mer complexes, a nucleant drop containing 0.05 mg/ml PEGylated graphene[35] was set up in addition to the control drop without the nucleant, and the two drops were equilibrated against the same reservoir particular to each condition. The best crystals were obtained in reservoir conditions containing 50 mM sodium cacodylate, 62.5 mM KCl, 7.5 mM MgCl$_2$, 2.5% Tacsimate™ (Hampton Research), 5.5–7% isopropanol, pH 6.5. Crystals were grown at 301°K until after 3-4 weeks, they reached their full size (Supplementary Table 4). The crystals were transferred into a cryoprotectant solution containing 50 mM Na-cacodylate (pH 6.5), 62.5 mM KCl, 7.5 mM MgCl$_2$, 2.5% Tacsimate™ (Hampton Research), 1 mM 2-mercaptoethanol and 30% (v/v) MPD, and then flash-frozen directly in liquid nitrogen.

## Data collection
X-ray diffraction data were collected at the Diamond Light Source synchrotron (Oxfordshire, UK) on beamline I04 equipped with the Eiger2 XE 16 M detector (wavelength of 0.95373 Å). The data were automatically processed by Xia2/DIALS[62].

For the topo IV-V18-delafloxacin complex grown in the presence of nucleant (8QMB), the unscaled integrated data from Xia2/DIALS[62] from 5 datasets from a single crystal were combined; while for the topo IV-V18-delafloxacin complex grown in the absence of nucleant (8QMC), the unscaled integrated data from a total of 8 datasets from 5 crystals were combined. In the case of the topo IV-E18-delafloxacin complex, the unscaled integrated data from 3 datasets from 3 separate crystals were combined. AIMLESS in the CCP4i2 suite was used to combine the isomorphous unscaled data[63]. As the datasets showed anisotropy, the unmerged data were further processed with STARANISO[64]. The X-ray data collection statistics are provided in Supplementary Tables 1 and 5–7.

## Structure solution and refinement
The X-ray crystal structures were solved by molecular replacement with PHASER in the CCP4i2 suite using the topo IV-DNA-levofloxacin complex pdb:3RAE as the search model[63,65]. The dictionary restraints for the delafloxacin molecule were generated by AceDRG using the smiles string C1C(CN1C2 = C(C = C3C(=C2Cl)N(C = C(C3 = O)C( = O)O)C4 = C(C = C(C( = N4)N)F)F)F)O[66]. The model was further refined using Refmac5 with several rounds of manual correction in Coot[67,68].

Towards the end of refinement, the structure was refined with PDB-REDO[69] as an additional check. The geometry of the structural models was validated using the tools in Coot and the PDB validation server. The refinement statistics are provided in Supplementary Table 1. The refined structures were deposited in the Protein Data Bank (PDB) under access codes 8QMB, 8QMC and 8C41.

Figures of proteins, DNA and drug structures were generated using UCSF ChimeraX[70].

## DNA cleavage assays
Our protocol for assaying DNA cleavage has been described previously[60]. Supercoiled pBR322 (0.4 μg) was incubated with full-length topo IV (reconstituted from 0.45 μg ParC and 1 μg ParE), with gyrase (0.45 μg GyrA plus 1 μg GyrB), or with ParE30-ParC55 fusion protein (0.4 μg) in reaction buffer (40 mM Tris-HCl, pH 7.5, 6 mM MgCl$_2$, 10 mM DTT, 200 mM potassium glutamate and 50 mg/ml BSA) in the absence or presence of delafloxacin (final volume 20 μl). After incubation for 1 h at 37 °C, 2 μl of 10% sodium dodecyl sulphate was added to each reaction to denature the proteins and release cleaved DNA. Finally, proteinase K was added to 200 μg/ml and incubation was

continued at 42 °C for 1 hr to remove DNA-bound protein. DNA products were separated by electrophoresis in a 1% agarose gel in TBE buffer. DNA was then visualised by staining the gel with ethidium bromide and photographed under UV illumination.

## Computational analysis of free drug conformation
Computational chemistry methods are now widely used in the pharmaceutical industry and elsewhere as a reliable means of small molecule structure prediction, often in lieu of X-ray crystal structure determination. In the absence of an X-ray crystal structure of delafloxacin, we used state-of-the-art quantum chemistry methods to predict its structure with high confidence. The molecular framework of delafloxacin was constructed using GaussView 6 and minimised first using the rapid semi-empirical PM7 method with water as a continuum solvent model[38]. This was followed by re-optimisation using the Gaussian 16 programme and employing the ωB97X-D density functional and the double-ζ Def2-SVP basis set, and then finally raising the level to the Def2-TZVPP triple-ζ basis set[39,40] with an SCRF=water level solvation method. The resulting structure revealed a tilted N-1 aromatic ring in accord with that seen in the free drug X-ray crystal structures of a closely related delafloxacin derivative (identical except for a slightly modified N-7 azetidinyl substituent) and of the structurally related fluoroquinolones trovafloxacin and tosufloxacin[37,51]. The predicted free delafloxacin structure closely overlaps the drug conformation seen here in the topo IV cleavage complex.

## Reporting summary
Further information on research design is available in the Nature Portfolio Reporting Summary linked to this article.

## Data availability
The refined X-ray crystal structures generated in this study have been deposited in the Protein Data Bank (PDB) under accession codes 8QMB, 8QMC and 8C41. The computational chemistry data generated in this study have been deposited in the Imperial College Research FAIR Data Repository (open access) and can be accessed from DOI: 10.14469/hpc/13324 [https://doi.org/10.14469/hpc/13324]. Source data are provided with this paper.

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

## Acknowledgements

We thank Dr Ralf Flaig and the IO4 beamline staff at the Diamond Synchrotron for help with data collection and processing, and Prof Mike Robb and Prof Mike Bearpark for their assistance in running Gaussian on the Imperial Computer Cluster. This research was supported by the Medical Research Council grant MR/T000848/1 (to L.M.F. and M.R.S.) and St George's, University of London.

## Author contributions

Conceived and designed the experiments: B.W., H.S.R., L.M.F., M.R.S. Performed the experiments: S.N., X.S.P., B.W., H.S.R., M.R.S. Analysed and interpreted the data: S.N., X.S.P., B.W., H.S.R., L.M.F., M.R.S. Contributed reagents/materials/analysis tools: X.S.P., B.W., L.G., N.E.C., N.R., M.S.P.S., H.S.R., L.M.F., M.R.S. Wrote the paper and critiqued the output for intellectual content prior to publication: L.M.F., S.N., X.S.P., B.W., L.G., N.E.C., N.R., M.S.P.S., H.S.R., M.R.S.

## Competing interests

The authors declare no competing interests.
