## [Transparent Peer Review file · Nature Communications]

Structural basis of topoisomerase targeting by delafloxacin

Corresponding Author: Professor Mark Sanderson

Version 0:

Reviewer comments:

Reviewer #1

(Remarks to the Author)

The manuscript by Najmudin et al. reports high-resolution crystal structures of delafloxacin-stabilized DNA cleavage complexes of *Streptococcus pneumoniae* topo IV. At 2.0 and 2.4 Å resolution, these structures reveal the 6-amino-3,5-difluoro-2-pyridinyl substituent at N-1 and chlorine at C-8 of the two Mg²⁺-chelated delafloxacin molecules intercalating at the DNA cleavage site in an unusual out-of-plane conformation. This configuration appears to enhance the binding affinity of delafloxacin to *S. pneumoniae* topo IV, which may be a key factor in its ability to counteract resistance to regular fluoroquinolones. This finding is medically important due to the unique role of delafloxacin in overcoming bacterial resistance to other fluoroquinolones. Additionally, the visualization of Mg²⁺ water bridges in these structures, which engage the enzyme, DNA, and drug, advances our understanding of the DNA cleavage/religation mechanism of topo IV and fluoroquinolone action. Furthermore, the discovery of a new K⁺ binding site in the topo IV complex highlights the importance of potassium ions in stabilizing the cleavage complex, providing another layer of understanding of the enzyme's functionality. Overall, this study offers a comprehensive view of delafloxacin's mechanism of action, presenting novel insights into fluoroquinolone interactions with bacterial topoisomerases. The reported findings not only enhance our understanding of delafloxacin's efficacy and suggest new strategies for designing antibiotics, but also provide new insights into type II topoisomerase mechanisms in general. I feel it is a valuable addition to Nature Communications. Nevertheless, I'd like to raise the following points for the authors to consider:

(1) It is intriguing that, despite belonging to the aromatic system, the N1 6-amino-3,5-difluoro-2-pyridinyl and C-8 chlorine substituents of delafloxacin adopt unusual out-of-plane configurations in the reported crystal structures. The stand-alone delafloxacin structures derived from QM calculations are consistent with these out-of-plane features. However, it would be more convincing if the authors could cite additional experimentally determined structures (besides delafloxacin or its analogs) that show steric effects causing bulky aromatic substituents to deviate from the typical sp² planar configuration.

(2) Given the potential significance of K⁺ to topo IV function, the conservation of residues involved in K⁺ binding should be analyzed and discussed. Since potassium glutamate is frequently used in analyzing the activities of eukaryotic type II topoisomerase, it would be interesting to determine if the K⁺ binding site is also present across eukaryotic enzymes. Additionally, as a key novelty of this work, it may be worthwhile to test the proposed functional significance of the K⁺ binding site through mutagenesis studies.

(3) A key finding of this work is that delafloxacin targets topo IV more effectively than DNA gyrase. Understanding the structural basis for this differential targeting would be valuable. This could be achieved through structural comparisons or by modeling the structure of the delafloxacin-stabilized gyrase cleavage complex using AlphaFold 3.

(4) With the high-resolution structures in hand, the authors are encouraged to speculate whether topo IV catalyzes DNA cleavage and religation via the "single dynamic Mg²⁺ model" or a variant of the "canonical two-metal dependent catalysis." The review paper titled "DNA Topoisomerase Inhibitors: Trapping a DNA-Cleaving Machine in Motion" by Bax et al. (*J. Mol. Biol.* 431:3427-3449 (2019)) may provide a useful foundation for this discussion.

Minor issues:

- (1) The Method section should describe exactly how graphene nucleant was used to improve crystal quality.
- (2) Compositions of the asymmetric units for both crystal forms should be provided so readers know if the two halves of the enzymes are related by crystallographic or non-crystallographic 2-fold symmetry.
- (3) Page 4, line 95: Would "passing" be a better word choice than "crossing"?

- (4) Page 6, line 195: Should it be “Protein binding induces a U-shaped bend” or “DNA binding induces a U-shaped bend”?
- (5) Page 9, line 289: Is it “the 2.0 Å...” or “the 2.4 Å...”?
- (6) Page 9, line 306: “manuscript in preparation”.... please consult the editor to see if this is acceptable for Nature Communications.
- (7) Page 10, line 320: “unpublished work”.... please consult the editor to see if this is acceptable for Nature Communications.

Reviewer #2

(Remarks to the Author)

Summary:

In this manuscript, the authors present the crystal structure of a *S. pneumo* TIV-delafloracin-DNA cleavage complex at the highest resolution reported to date. All data presented regarding the well-known “water-metal-ion bridge” (i.e., keto acid portion of the drug, the chelated Mg²⁺ ion, its hydration sphere, and coordination to the enzyme via S79 and D/E83) are consistent with previously published structural and functional work. The main interest in this paper lies with the drug being used, as delafloracin, despite being a FQ, seems to overcome the most common resistance mutations that render FQs commonly used in the clinic, such as cipro and levo, ineffective. The structure provided here suggests multiple interactions between the substituents of delafloracin that make it unique (as compared to other FQs) that could explain how it overcomes resistance and apparently maintains enzyme interaction when the anchors of the water-metal-ion bridge are mutated. Some of these interactions appear to involve bridging interactions, which involve metal ions and/or water molecules, to the DNA or to enzyme residues other than S79 and D/E83.

Specific comments:

Throughout, including Lines 237-255, 327-344, 403-420, and 428-449: The water-metal ion bridge that serves as the main interaction point between FQs and type II topoisomerases is well-established in the literature as consisting of the Mg²⁺ chelated by the keto acid of the drug and then linking to the enzyme via water molecules with the enzyme anchors being S79 and D/E83. Here, the authors seem to try to re-define this model as being multiple bridges. This is inappropriate because this model is already well-established and the structure presented here does not suggest any alterations to this established model. Thus, the authors should re-write these sections to reflect what is already established and accepted. For the “new” bridges that the authors are suggesting here, such as those to the DNA, they can count those how they see fit. The bridge counts discussed are currently confusing, and re-writing these sections to reflect the previously established model mentioned above will likely help clarify, but it may also be beneficial to provide a basic, summary illustration (similar to those seen in PMID 23460203 and PMID 28708938) of all of the “new” bridges.

Abstract, line 38: Rephrase this, as it currently implies that K⁺ ions participate in the drug interaction – the manuscript indicates they are likely structural supports for the enzyme and interact far away from the drug.

Figure 1: Include ciprofloxacin, as it is mentioned a number of times throughout the manuscript and is generally regarded as the benchmark FQ to which things are compared. In addition, switch the places of moxi and clina so that the clinical FQs are together and the experimental are together at the bottom.

Figure 2: In the caption, note which steps are cleavage complexes, as not all readers may be well-versed in topo function.

Line 100: aspartate should be changed to acidic, as some enzymes have E instead of D at this position

Line 109: Was this reference intended to be for Figure 2c rather than 2b?

Lines 111-113: Delete this sentence as it is highly repetitive with the following sentence.

Lines 120-122: Include a reference for this clinical trial and its outcome(s).

Lines 130-131: After reading the manuscript, it is not clear what “longstanding mechanistic issues general to all FQs” is referring to.

Figure 3: The caption needs to state: how many experiments these gels are representative of; the type of gel (TAE or TBE) used as well as whether EtBr was included in the gel or whether the gel was stained afterward; and how many ug of enzyme were used in panel b. (The relevant part of the Methods section should also include this missing information.) In addition to the gels, include a quantification of all gels these experiments represent since fold comparisons are discussed in the results. Lastly, were the experiments in panel b really carried out in the absence of ATP? This looks more like a catalytic assay than a cleavage assay given the relaxation observed, and to my knowledge, topo IV cannot carry out its catalytic activity without ATP – ATP is not listed as being part of these reactions. (Regarding this, Line 154 says this is a “well-known” activity, but only cites one reference, so this language would be more accurate to say a “previously known” activity.)

Line 157: The parenthetical phrase is not entirely clear. I suspect the authors mean DSB in comparison to SSB. This should be clarified. In addition, the authors may want to mention in the previous paragraph the high levels of SSB seen with this FQ, especially with topo IV. (And they may want to even include multiple quantifications for comparison – one for SSB, one for DSB, and one for the “total” cleavage of DSB and SSB combined.)

Lines 184-186: Reference 29 should be included here as it is the first paper to capture the Mg²⁺ ion found to function in the water-metal ion bridge interaction between the drug and S79 and D/E83, and this manuscript is now the second to capture and image this key metal ion.

Figure 4: It does not appear that the potassium ions in the bottom half of the figure are circled as indicated in the caption. Circling them in a contrasting color in the bottom half would be beneficial as they are very hard to find.

Line 231: Figure 9 is cited before Figure 8, so these two figures should be reversed.

Figure 9: Caption states that S79 and D83 are in red, which does not appear to be the case.

Line 248-249: “structural” should be added between “unequivocal” and “evidence”, as structures and their implications are hypothetical until there is functional biochemical evidence that supports them. To this point, at minimum, PMID 23460203 should be cited here, though the authors may wish to also include the other papers that have provided functional evidence for this interaction (which I believe that list would be completed by PMID 25115926, PMID 26792518, PMID 28708938, and PMID 36769202.)

Lines 281-286: The two-metal-ion mechanism has also been tested and confirmed in *E. coli* topo IV. This finding, along with the appropriate reference (PMID 21300644), should also be included here. Furthermore (and as mentioned in the point immediately preceding this one), given that the hopping metal ion mechanism has been disproven with functional data, it is not clear why this section is included. (This does not really have anything to do with delafloxacin and how it potentially overcomes resistance, which seems to be the main point/impactful finding of this work.)

Lines 311-313: Address the likelihood of these K⁺ sites existing and being accessible in the full enzyme (i.e., one composed of full length A and B subunits).

Figure 10: The dashed lines in the bottom half of the figure appear to be black, rather than the dark blue stated in the caption.

Lines 317-318: “It is known that *E. coli* gyrase requires K⁺ for supercoiling activity” – provide references for this statement.

Lines 318-319: Glutamate is well-known to enhance interaction between DNA and many types of enzymes, including polymerases, by essentially making the DNA “sticky”. Thus, it is unlikely that the counter ion makes a difference. Unless the authors want to repeat these experiments using a different glutamate salt, such as sodium glutamate, this should be removed as it really does not support their K⁺ ion argument.

Line 320: It is not clear why Ref 19 is included here as it does not address the necessity of K⁺ ions.

Line 323: Change “in” to “on” since these ions are on the surface of the enzyme.

Line 337: Clarify that the out-of-plane portion is 2 of the substituents, not the entire drug molecule.

Line 343: change “a” FQ to “this” FQ because many of the interactions detailed in this paper are specific to delafloxacin.

Lines 357-358: Provide references for this statement.

Lines 373-374: Provide references for this statement.

Line 384: Provide a couple of examples of “other unrelated drugs”

Line 396: It is not clear why Ref 27 is included here.

Line 419: Figure 12 is cited but not provided.

Line 431-433: “Mutations at these loci...to trap the cleavage complex.” This statement should be referenced (see PMIDs of appropriate papers in the point above regarding Lines 248-249). Additionally, this statement is not accurate. For both *E. coli* enzymes (topo IV and gyrase), the bridge was reported to play a positioning, rather than binding, role, so this should be corrected.

Line 446: The language should be changed here from “direct and convincing evidence” to “direct visual evidence” because, as noted above in the note on Lines 248-249, functional evidence is key.

Line 449: The primary literature papers examining function of the bridge (see PMIDs in the point above regarding Lines 248-249) should be cited here.

Lines 458-461: "Presumably, ...ParC/GyrA." The point of this statement and how it fits with the previous statement(s) is unclear.

Line 480: Was Figure 3, not Figure 4, supposed to be referenced here?

Line 482: Weakening would probably be a better word choice than loss here, as some of the functional papers addressing the water-metal ion bridge indicated that either residue could coordinate the bridge but that it was strongest when both were present.

Supplementary Figure 1: Remove trovafloxacin, as it is already included in Figure 1.

Supplementary Figure 4 does not appear to be cited anywhere in the manuscript.

Reviewer #3

(Remarks to the Author)

This paper describes the crystal structure of the complex between a large fragment of *Streptococcus pneumoniae* DNA topoisomerase IV, DNA and the fluoroquinolone (FQ) drug delafloxacin. The ms is well-written and the figures are clear. FQs have been highly successful antibiotics, and delafloxacin is approved for various pneumococcal infections. As such, this is an important area that is potentially of interest to scientists from various disciplines. However, many structures of topoisomerase-DNA-FQ complexes have been solved, so the question arises as to how significant this ms is. Firstly, *S. pneumoniae* is an important pathogen; secondly delafloxacin is an unusual FQ and thirdly the resolution of this complex is high, allowing more information to be gleaned from the structure. However, despite these points I am not sure that this ms should be published in Nature comms., as I feel it lacks impact and significance. There are several points to be drawn to the authors' attention:

- i. The structure is solved with the truncated form of the enzyme: ParE30-ParC55. It would be better if the other parts of the enzyme were present, as in the recent Cryo-EM structures (1,2), one of which contains moxifloxacin.
- ii. There are quite a lot of figures in the main text and I wonder if some of these should be moved to the Supplementary Materials.
- iii. P. 13, line 435: (26) is the wrong reference here.
- iv. Although ms is well-written there are a few places where hyphens are missing (e.g. 2.4-Å resolution), these would improve clarity.

1. Michalczyk, E., Pabiś, M., Heddle, J. and Ghilarov, D. (2024) Structure of *Escherichia coli* DNA gyrase with chirally wrapped DNA supports ratchet-and-pawl mechanism for an ATP-powered supercoiling motor. bioRxiv, doi: <https://doi.org/10.1101/2024.04.12.589215>.

2. Vayssieres, M., Marechal, N., Yun, L., Lopez Duran, B., Murugasamy, N.K., Fogg, J.M., Zechiedrich, L., Nadal, M. and Lamour, V. (2024) Structural basis of DNA crossover capture by *Escherichia coli* DNA gyrase. *Science*, 384, 227-232.

Version 1:

Reviewer comments:

Reviewer #1

(Remarks to the Author)

The authors' responses to my suggestions are adequate, and I have no further concerns. I recommend the publication of this manuscript.

Reviewer #2

(Remarks to the Author)

Prior comments/concerns on the original draft have been sufficiently and satisfactorily addressed in this revised manuscript.

Reviewer #3

(Remarks to the Author)

The authors have addressed my comments satisfactorily.

Reviewer #4

(Remarks to the Author)

My area of expertise related to this paper is only on the quantum mechanical calculations. Therefore, I have only a few questions regarding the computational details:

The authors stated that: The molecular framework of delafloxacin was constructed using GaussView6 and minimised first using PM7 with water as solvent. This was followed by running Gaussian 16 using the wB97XD/ Def2tzvpp basis set and then using a wB97XD/Def2-SVPP/SCRF=water level basis set.

The computational details are not correctly described. 1) First, what the authors refer with running Gaussian16? Did they perform a geometry optimization?

2) Second, wB97XD/ Def2tzvpp is a level of theory not just a basis set. The wB97XD is the DFT functional used and the Def2tzvpp is the basis set.

3) The second part, "then using a wB97XD/Def2-SVPP/SCRF=water level basis set." is also not properly explained. This calculation is a single point energy calculation based on the optimized structure?

4) The wB97XD/Def2-SVPP/SCRF=water is again a level of theory, not a "level basis set."

I can not recommend the paper for publication without these issues being addressed.

Version 2:

Reviewer comments:

Reviewer #4

(Remarks to the Author)

My questions have been addressed in the current version. I have no further questions.

Manuscript NCOMMS-24-4016

'Structural basis of topoisomerase targeting by delafloxacin, a novel anionic fluoroquinolone'

Authors: Najmudin S. et al

Response to Reviewers' comments

(Changes in the revised manuscript are shown in blue font).

REVIEWER COMMENTS

Reviewer #1 (Remarks to the Author): *Point-by-point response*

The manuscript by Najmudin et al. reports high-resolution crystal structures of delafloxacin-stabilized DNA cleavage complexes of *Streptococcus pneumoniae* topo IV. At 2.0 and 2.4 Å resolution, these structures reveal the 6-amino-3,5-difluoro-2-pyridinyl substituent at N-1 and chlorine at C-8 of the two Mg²⁺-chelated delafloxacin molecules intercalating at the DNA cleavage site in an unusual out-of-plane conformation. This configuration appears to enhance the binding affinity of delafloxacin to *S. pneumoniae* topo IV, which may be a key factor in its ability to counteract resistance to regular fluoroquinolones. This finding is medically important due to the unique role of delafloxacin in overcoming bacterial resistance to other fluoroquinolones. Additionally, the visualization of Mg²⁺ water bridges in these structures, which engage the enzyme, DNA, and drug, advances our understanding of the DNA cleavage/religation mechanism of topo IV and fluoroquinolone action. Furthermore, the discovery of a new K⁺ binding site in the topo IV complex highlights the importance of potassium ions in stabilizing the cleavage complex, providing another layer of understanding of the enzyme's functionality. Overall, this study offers a comprehensive view of delafloxacin's mechanism of action, presenting novel insights into fluoroquinolone interactions with bacterial topoisomerases. The reported findings not only enhance our understanding of delafloxacin's efficacy and suggest new strategies for designing antibiotics, but also provide new insights into type II topoisomerase mechanisms in general. I feel it is a valuable addition to Nature Communications. Nevertheless, I'd like to raise the following points for the authors to consider:

(1) It is intriguing that, despite belonging to the aromatic system, the N1 6-amino-3,5-difluoro-2-pyridinyl and C-8 chlorine substituents of delafloxacin adopt unusual out-of-plane configurations in the reported crystal structures. The stand-alone delafloxacin structures derived from QM calculations are consistent with these out-of-plane features. However, it would be more convincing if the authors could cite additional experimentally determined structures (besides delafloxacin or its analogs) that show steric effects causing bulky aromatic substituents to deviate from the typical sp² planar configuration.

In our submitted manuscript, we cited references to several experimentally determined structures showing bulky aromatic substituents that are tilted i.e. deviate from typical sp^2 planar geometry. These include a crystal structure of a C-7 substituted delafloxacin molecule (Kuramoto et al, 2003) and crystal structures of trovafloxacin and tosufloxacin (Brighty and Gootz, 1997). These are not 'analogs' of delafloxacin and as we discuss at length, trovafloxacin like delafloxacin also has a tilted N-1 aromatic ring in both the crystal structure of the free drug and the topo IV-bound state (our PDB structures). To address how common is this deviation from sp^2 geometry, we include the results of a search we did of the Cambridge small molecule database using a generic quinolone framework with an aromatic substituent at N-1. (See new Supplementary Fig 6). As we describe in the revised Discussion (new paragraph 4), 178 entries were found with some structures having a near planar N-1 aromatic ring whereas in others it is tilted. So delafloxacin is not unique in having a non- sp^2 geometry. To note, our studies of delafloxacin are self-standing and not reliant on other structures.

(2) Given the potential significance of K^+ to topo IV function, the conservation of residues involved in K^+ binding should be analyzed and discussed. Since potassium glutamate is frequently used in analyzing the activities of eukaryotic type II topoisomerase, it would be interesting to determine if the K^+ binding site is also present across eukaryotic enzymes. Additionally, as a key novelty of this work, it may be worthwhile to test the proposed functional significance of the K^+ binding site through mutagenesis studies.

We now include analysis showing that X-ray/cryoEM structures of bacterial and eukaryotic type II topoisomerases are overall highly conserved (new Supplementary Table 2). Moreover, in new Supplementary Fig. 7, we show there is strong sequence and structural conservation of the K^+ binding site found in pneumococcal topo IV across bacterial topo IV and gyrase enzymes (Suppl Fig.7a, 7b). The equivalent sequences in eukaryotic type II enzymes show a different sequence conservation (Suppl Fig 7a), though they all form a similar alpha helix-loop-beta sheet fold to that of the bacterial enzymes (Fig. 7c). Thus, sequence and structural conservation suggest a functional role for the K^+ binding site across Top2A enzymes.

Moreover, for the bacterial topoisomerase complexes bound to a sufficiently long DNA, the potassium ion is in range (3-4 Å) to bind a phosphate oxygen of DNA. The same appears to be true for the eukaryotic Top2As though the putative bond to the DNA phosphate oxygen is slightly longer. This additional analysis suggests a potential role for K^+ in binding and perhaps bending the gate-DNA is described in the Discussion, new paragraphs 8 and 9.

We agree with the Reviewer that it would be worthwhile to test the functionality of the K^+ binding site using mutagenesis approaches. However, as the K^+ binding pocket is mostly supported by protein backbones, a future mutagenesis study would need careful thought and may be difficult to design. Mutagenesis is never about a single substitution. Multiple changes would need to be introduced and screened which would require a substantial new project in its own right. Consequently, we have in paragraph 9 included a caveat by stating 'Clearly more work will be needed to ascertain whether other Top2As contain K^+ and to establish its proposed functional and structural roles using site-directed mutagenesis'.

(3) A key finding of this work is that delafloxacin targets topo IV more effectively than DNA gyrase. Understanding the structural basis for this differential targeting would be valuable. This could be achieved through structural comparisons or by modelling the structure of the delafloxacin-stabilized gyrase cleavage complex using AlphaFold 3.

To understand the differential targeting of topo IV over gyrase, a structural comparison of delafloxacin arrested DNA cleavage complexes of topo IV and gyrase would be worthwhile. However, we note it took more than a decade of effort for us to obtain the high-resolution X-ray structures of topo IV-delafloxacin cleavage complexes reported here and we do not have the equivalent gyrase structure for comparison as its DNA requirements are different and not fully understood. An alternative would be to use AlphaFold 3 to predict the structure of the delafloxacin stabilized gyrase cleavage complex but the programs are not yet well developed for predicting a structure that involves multiple interacting proteins, multiple and different metal ions, drugs, DNA and very many participating water molecules (whose locations in gyrase are currently unknown). Consequently, this suggestion is valid but poses significant challenges that in our view are best resolved by future efforts to obtain a high-resolution X-ray structure of the equivalent pneumococcal gyrase complex, possibly nucleant assisted.

Accordingly, we have added a sentence in explanation to the end of paragraph 2 of the Discussion.

Also, to note in relation to the present work, it is targeting in vivo that is more clinically relevant and that is determined not just by substrate affinity for gyrase/topo IV but also by other factors including the differing intracellular target levels as well as target lethality.

(4) With the high-resolution structures in hand, the authors are encouraged to speculate whether topo IV catalyzes DNA cleavage and religation via the "single dynamic Mg²⁺ model" or a variant of the "canonical two-metal dependent catalysis." The review paper titled "DNA Topoisomerase Inhibitors: Trapping a DNA-Cleaving Machine in Motion" by Bax et al. (J. Mol. Biol. 431:3427-3449 (2019)) may provide a useful foundation for this discussion.

In common with earlier low-resolution structures of core cleavage complexes with quinolones, our high-resolution structures reported here show the presence of a single pair of 'active site' Mg²⁺ ions (one per cleaved DNA strand) located at the 'B site' stabilising the cleaved DNA. Therefore, the present work per se does not provide fresh insight on the 'one' versus 'two metal ion' mechanism question. Moreover, Reviewer 2 noted that our work is largely about delafloxacin, and asked that we remove our previous discussion on the mechanism. We have chosen to mention the mechanistic uncertainty and have cited all the relevant papers, including the Bax et al review as new reference 54. See also response to Referee 2 re lines 281-286.

Minor issues:

(1) The Method section should describe exactly how graphene nucleant was used to improve crystal quality.

Use of the graphene nucleant is now described in paragraph 2 of the Crystallization section of the Methods.

(2) Compositions of the asymmetric units for both crystal forms should be provided so readers know if the two halves of the enzymes are related by crystallographic or non-crystallographic 2-fold symmetry.

In regard, to asymmetric unit composition, we now state on in the 'Canonical features' section of the Results that that the two halves of the enzyme structure are related by non-crystallographic symmetry, with the ParE30-ParC55 chain as the asymmetric subunit.

(3) Page 4, line 95: Would "passing" be a better word choice than "crossing"?

Agreed. Text altered accordingly.

(4) Page 6, line 195: Should it be "Protein binding induces a U-shaped bend" or "DNA binding induces a U-shaped bend"?

It should be the former. Text altered accordingly.

(5) Page 9, line 289: Is it "the 2.0 Å..." or "the 2.4 Å..."?

Correct as written-the 2.0 Å delafloxacin structure best images the 'B site' Mg²⁺ ions.

(6) Page 9, line 306: "manuscript in preparation".... please consult the editor to see if this is acceptable for Nature Communications.

This paper has been published in PNAS and is now included as new reference 57.

(7) Page 10, line 320: "unpublished work".... please consult the editor to see if this is acceptable for Nature Communications.

In response to Reviewer #2, mention of potassium glutamate has now been removed.

Therefore, no need to cite 'unpublished work'.

Reviewer #2 (Remarks to the Author):

Summary:

In this manuscript, the authors present the crystal structure of a *S. pneumo* TIV-delafloxacin-DNA cleavage complex at the highest resolution reported to date. All data presented regarding the well-known "water-metal-ion bridge" (i.e., keto acid portion of the drug, the chelated Mg²⁺ ion, its hydration sphere, and coordination to the enzyme via S79 and D/E83) are consistent with previously published structural and functional work. The main interest in this paper lies with the drug being used, as delafloxacin, despite being a FQ, seems to overcome the most common resistance mutations that render FQs commonly used in the clinic, such as cipro and levo, ineffective. The structure provided here suggests

multiple interactions between the substituents of delafloxacin that make it unique (as compared to other FQs) that could explain how it overcomes resistance and apparently maintains enzyme interaction when the anchors of the water-metal-ion bridge are mutated. Some of these interactions appear to involve bridging interactions, which involve metal ions and/or water molecules, to the DNA or to enzyme residues other than S79 and D/E83.

Specific comments:

Throughout, including Lines 237-255, 327-344, 403-420, and 428-449: The water-metal ion bridge that serves as the main interaction point between FQs and type II topoisomerases is well-established in the literature as consisting of the Mg^{2+} chelated by the keto acid of the drug and then linking to the enzyme via water molecules with the enzyme anchors being S79 and D/E83. Here, the authors seem to try to re-define this model as being multiple bridges. This is inappropriate because this model is already well-established, and the structure presented here does not suggest any alterations to this established model. Thus, the authors should re-write these sections to reflect what is already established and accepted. For the "new" bridges that the authors are suggesting here, such as those to the DNA, they can count those how they see fit. The bridge counts discussed are currently confusing, and re-writing these sections to reflect the previously established model mentioned above will likely help clarify, but it may also be beneficial to provide a basic, summary illustration (similar to those seen in PMID 23460203 and PMID 28708938) of all of the "new" bridges.

We recognise that the term 'water-metal ion bridge' has an established usage in describing the binding between the quinolone-chelated Mg^{2+} ion and one or both conserved enzyme ParC S79 and D/E83 residues. The term is conveniently imprecise allowing discussion of biochemical findings in the absence of structural information as is sometimes the case. We accept the argument that a precedent has been set and accordingly we have rewritten the text throughout restricting the use of 'water-metal ion bridge' to the links between the drug bound- Mg^{2+} and one or both ParC residues. We refer to all the other newly visualised long-range interactions between Mg^{2+} ions and DNA/protein via water as 'water-mediated links.' As to the Reviewer's last point regarding inclusion of an illustration, clear schematic diagrams (Fig. 7 and 10) are already provided showing the full coordination of Mg^{2+} ions directly or via molecules to DNA and protein .

Abstract, line 38: Rephrase this, as it currently implies that K^+ ions participate in the drug interaction – the manuscript indicates they are likely structural supports for the enzyme and far away from the drug.

Agreed. Abstract has been amended.

Figure 1: Include ciprofloxacin, as it is mentioned a number of times throughout the manuscript and is generally regarded as the benchmark FQ to which things are compared.

In addition, switch the places of moxi and clina so that the clinical FQs are together and the experimental are together at the bottom.

Agreed. We have reorganised Figure 1 exactly as suggested to include the structure of ciprofloxacin. Legend updated.

Figure 2: In the caption, note which steps are cleavage complexes, as not all readers may be well-versed in topo function.

Clarifying sentence has been added to the Fig. 2 caption.

Line 100: aspartate should be changed to acidic, as some enzymes have E instead of D at this position.

Now corrected in the text.

Line 109: Was this reference intended to be for Figure 2c rather than 2b?

Yes. Now corrected to read Fig. 2c.

Lines 111-113: Delete this sentence as it is highly repetitive with the following sentence.

Agreed. Sentence deleted.

Lines 120-122: Include a reference for this clinical trial and its outcome(s).

The review by Collins and Osheroff (ref 2) incorporating five references to clinical trials for gepotidacin and zoliflodacin is now cited.

Lines 130-131: After reading the manuscript, it is not clear what “longstanding mechanistic issues general to all FQs” is referring to.

Sentence has been rewritten to state... ‘In combination with biochemical and computational studies, our work provides new insights on how delafloxacin binds the cleavage complex to retain clinical activity against resistant strains, an important paradigm in designing drugs to counter antimicrobial resistance’.

Figure 3: The caption needs to state: how many experiments these gels are representative of; the type of gel (TAE or TBE) used as well as whether EtBr was included in the gel or whether the gel was stained afterward; and how many ug of enzyme were used in panel b. (The relevant part of the Methods section should also include this missing information.) In addition to the gels, include a quantification of all gels these experiments represent since fold comparisons are discussed in the results. Lastly, were the experiments in panel b really carried out in the absence of ATP? This looks more like a catalytic assay than a cleavage assay given the relaxation observed... and to my knowledge, topo IV cannot carry out its catalytic activity without ATP – ATP is not listed as being part of these reactions. (Regarding this, Line 154 says this is a “well-known” activity, but only sites one reference, so this language would be more accurate to say a “previously known” activity.)

Several points are raised here. First, the experiments in Fig. 3a were conducted once primarily to aid structural studies (rather than provide inhibition parameters to decimal place precision). Experiments were carefully done and show that as for other quinolones, delafloxacin captures a cleavage complex more efficiently with topo IV than gyrase, a great advantage in structural work. Moreover, the difference in target sensitivity is marked and we can reasonably point to a 10-20-fold greater sensitivity for topo IV. The studies in Fig. 3b were to check the cleavage activity of our topo IV fusion protein. Delafloxacin is comparably active against the topo IV fusion and holoenzyme complexes (Fig. 3a and b). Finally, as we now mention in the text, the Fig. 3b results for levofloxacin recapitulate those we previously published in Laponogov et al, 2016 using the same conditions and levels of drug, fusion protein and pBR322 DNA thereby demonstrating a consistency of approach.

These additional points are now covered in the revised Results section on 'Capture of topo IV and gyrase cleavage complexes.'

Fig. 3 caption and Materials and Methods. In the revised version, we now include mention of the TBE gel buffer system and addition of ethidium bromide stain after electrophoresis. In panel b, the level of fusion protein (0.4 microgram) was already included.

*Finally, our description of experiments in panel 3b using the ParE30-ParC55 fusion protein is correct as written. As we have reported earlier (ref 30), the fusion protein (that lacks an ATP site and β -pinwheel CTD) relaxes supercoiled plasmid DNA in the absence of ATP. Similarly, in earlier work on *E. coli* gyrase (Gellert M, Fisher LM and O'Dea, PNAS 1979) we showed that a fragment of Gyrase B protein (residues 394-804 comprising the TOPRIM domain but lacking the ATPase site [Adachi T et al, Nucleic Acids Res 1, 771-784 (1987)], also relaxed supercoiled pBR322 DNA when complemented with GyrA in the absence of ATP. This Gellert et al study is mentioned in the text and cited as new reference 41.*

Line 157: The parenthetical phrase is not entirely clear. I suspect the authors mean DSB in comparison to SSB. This should be clarified. In addition, the authors may want to mention in the previous paragraph the high levels of SSB seen with this FQ, especially with topo IV. (And they may want to even include multiple quantifications for comparison – one for SSB, one for DSB, and one for the "total" cleavage of DSB and SSB combined.)

Yes. We do mean DSB compared to SSB and have clarified the text (second para of the results) accordingly. In the previous paragraph, we now mention the high levels of single stranded breaks seen with delafloxacin, especially with topo IV.

Lines 184-186: Reference 29 should be included here as it is the first paper to capture the Mg²⁺ ion found to function in the water-metal ion bridge interaction between the drug and S79 and D/E83, and this manuscript is now the second to capture and image this key metal ion.

Agreed. Reference 29 now cited here in the text.

Figure 4: It does not appear that the potassium ions in the bottom half of the figure are circled as indicated in the caption. Circling them in a contrasting color in the bottom half would be beneficial as they are very hard to find.

As noted, potassium ions are circled clearly in the top half of Figure 4. The ions are completely obscured by other structures in the bottom right-hand projection but are distinctly visible in the bottom left-hand figure. Rather than circling the ions (which clashes with alpha helical segments) we have now identified them using black arrows.

Line 231: Figure 9 is cited before Figure 8, so these two figures should be reversed.

Citation of Figures 8 and 9 has been corrected in the text so they now appear in numerical order.

Figure 9: Caption states that S79 and D83 are in red, which does not appear to be the case.

Agreed. Caption now corrected.

Line 248-249: "structural" should be added between "unequivocal" and "evidence", as structures and their implications are hypothetical until there is functional biochemical evidence that supports them. To this point, at minimum, PMID 23460203 should be cited here, though the authors may wish to also include the other papers that have provided functional evidence for this interaction (which I believe that list would be completed by PMID 25115926, PMID 26792518, PMID 28708938, and PMID 36769202.)

The word 'structural' has been added as suggested and PMID 23460203 is now cited here (and elsewhere as new reference 51. The other 4 PMIDs each repeat much the same approach on closely related enzymes. Given we are already above the suggested 70 reference limit, we have cited the excellent review (ref 2) indicating the four other PMIDs as refs 162-165 therein.

Lines 281-286: The two-metal-ion mechanism has also been tested and confirmed in E. coli topo IV. This finding, along with the appropriate reference (PMID 21300644), should also be included here. Furthermore (and as mentioned in the point immediately preceding this one), given that the hopping metal ion mechanism has been disproven with functional data, it is not clear why this section is included. (This does not really have anything to do with delafloxacin and how it potentially overcomes resistance, which seems to be the main point/impactful finding of this work.)

There is still much uncertainty about whether type II topoisomerases act via a two or one metal ion mechanism (e.g. see Reviewer 1 point 4 and cited review). However, we agree with the Referee that our manuscript is primarily about delafloxacin and does not add to the mechanistic debate. Therefore, as suggested, we have deleted lines 281-286 and instead added a short general introduction to this section in which the relevant papers are cited including PMID 21300644-now new reference 53.

Lines 311-313: Address the likelihood of these K⁺ sites existing and being accessible in the full enzyme (i.e., one composed of full-length A and B subunits).

In the revised version, we now include new Supplementary Table 2 showing that overall, the crystal structures of prokaryotic and eukaryotic type II topoisomerases are highly conserved. Moreover, in new Supplementary Fig 7, we show that the sequence of the tower domain motif that binds K⁺ in pneumococcal topo IV is highly conserved with a similar 3-D fold in both bacterial and eukaryotic type II enzymes. Furthermore, the K⁺ is in range to bind a phosphate oxygen of the gate DNA. We note that the site is occupied (presumably by K⁺) in the structure of an S. pneumoniae topo IV-DNA complex comprising the full-length ParE and ParC55 breakage-reunion domain (PDB 4I3H). These aspects are now presented in new paragraphs 8 and 9 of the Discussion.

Figure 10: The dashed lines in the bottom half of the figure appear to be black, rather than the dark blue stated in the caption.

OK. Fig. 10 legend has been corrected.

Lines 317-318: "It is known that E. coli gyrase requires K⁺ for supercoiling activity" – provide references for this statement.

We have modified the text to state 'It is known that gyrase requires K⁺ for catalytic activity' and cited new references 58 and 59.

Lines 318-319: Glutamate is well-known to enhance interaction between DNA and many types of enzymes, including polymerases, by essentially making the DNA "sticky". Thus, it is unlikely that the counter ion makes a difference. Unless the authors want to repeat these experiments using a different glutamate salt, such as sodium glutamate, this should be removed as it really does not support their K⁺ ion argument.

The activating effect of potassium glutamate on bacterial topo IV and gyrase is just as likely due to K⁺. However, as we did not show this directly, we have removed mention from the text.

Line 320: It is not clear why Ref 19 is included here as it does not address the necessity of K⁺ ions.

Agreed. Ref 19 has been removed.

Line 323: Change "in" to "on" since these ions are on the surface of the enzyme.

We prefer 'in' over 'on'. The potassium ion occupies a pocket comprising an alpha helix, loop and beta strand; a structure we now show is highly conserved among Top2As (see additional text in the Discussion and new Supplementary Figure 7).

Line 337: Clarify that the out-of-plane portion is 2 of the substituents, not the entire drug molecule.

Thank you. Now clarified in the text.

Line 343: change "a" FQ to "this" FQ because many of the interactions detailed in this paper are specific to delafloxacin.

Agreed. Changed in the text.

Lines 357-358: Provide references for this statement.
Statement is now explained with new references 50 and 61 inserted.

Lines 373-374: Provide references for this statement.
Reference 1 has been added.

Line 384: Provide a couple of examples of "other unrelated drugs"
Two examples now included in the text.

Line 396: It is not clear why Ref 27 is included here.
Ref 27 has been corrected to read ref 28 (which describes the first crystal structure of a quinazolinone-stabilised cleavage complex).

Line 419: Figure 12 is cited but not provided.
Thank you. Corrected to read '(Fig. 10)'.

Line 431-433: "Mutations at these loci...to trap the cleavage complex." This statement should be referenced (see PMIDs of appropriate papers in the point above regarding Lines 248-249). Additionally, this statement is not accurate. For both E. coli enzymes (topo IV and gyrase), the bridge was reported to play a positioning, rather than binding, role, so this should be corrected.
Appropriate references have been included as before. The text has been corrected by adding a sentence to mention that depending on the enzyme and bacterial species, the water metal ion bridge can be disrupted to reduce drug affinity, whereas in some cases e.g. E. coli topo IV and gyrase, the bridge may have a drug positioning rather than drug binding role.

Line 446: The language should be changed here from "direct and convincing evidence" to "direct visual evidence" because, as noted above in the note on Lines 248-249, functional evidence is key.
We would rather keep the original wording as only structural studies can reveal the precise molecular nature of an interaction. We agree that biochemical studies are equally important as they provide complementary information on function.

Line 449: The primary literature papers examining function of the bridge (see PMIDs in the point above regarding Lines 248-249) should be cited here. *References have now been included.*

Lines 458-461: "Presumably, ...ParC/GyrA." The point of this statement and how it fits with the previous statement(s) is unclear.

Agreed. The two sentences in question have been revised to clarify the meaning.

Line 480: Was Figure 3, not Figure 4, supposed to be referenced here?

Yes. We have amended the text to read 'Fig. 3'.

Line 482: Weakening would probably be a better word choice than loss here, as some of the functional papers addressing the water-metal ion bridge indicated that either residue could coordinate the bridge but that it was strongest when both were present.

Text has been amended (line 483) as suggested.

Supplementary Figure 1: Remove trovafloxacin, as it is already included in Figure 1.

We have kept the structure of trovafloxacin in Supplementary Figure 1 as it spares the reader having to switch between the Main figures and Supplementary section when comparing quinolone structures.

Supplementary Figure 4 does not appear to be cited anywhere in the manuscript.

Now cited in line 235.

Reviewer #3 (Remarks to the Author):

This paper describes the crystal structure of the complex between a large fragment of *Streptococcus pneumoniae* DNA topoisomerase IV, DNA and the fluoroquinolone (FQ) drug delafloxacin. The ms is well-written and the figures are clear. FQs have been highly successful antibiotics, and delafloxacin is approved for various pneumococcal infections. As such, this is an important area that is potentially of interest to scientists from various disciplines. However, many structures of topoisomerase-DNA-FQ complexes have been solved, so the question arises as to how significant this ms is. Firstly, *S. pneumoniae* is an important pathogen; secondly delafloxacin is an unusual FQ and thirdly the resolution of this complex is high, allowing more information to be gleaned from the structure. However, despite these points I am not sure that this ms should be published in Nature comms., as I feel it lacks impact and significance.

The reviewer lists many strong reasons for publication in an overview that is only a partial summary. Omitted are the key points detailed by Reviewer 1 who describes the outcomes of our work that make it a 'valuable contribution to Nature Comms' including the unusual out of plane conformation of the drug, the precise roles of Mg²⁺ ions, the characterisation of new-found ParC binding sites for potassium ions, and importantly the impact and medical significance of understanding how delafloxacin can overcome clinical resistance to other quinolones- a valuable asset at a time when drug resistance is a growing medical problem. (The reviewer could also have highlighted the first use of a graphene nucleant to obtain a 2Å structure, a resolution unprecedented for a quinolone-cleavage complex). Clearly, as we now emphasize at the end of the Introduction, delafloxacin is a drug with unique properties that make it stand out from other fluoroquinolones. Our multidisciplinary study will be of broad

scientific and medical interest and as such we believe the work is appropriate for Nature Communications.

There are several points to be drawn to the authors' attention:

i. The structure is solved with the truncated form of the enzyme: ParE30-ParC55. It would be better if the other parts of the enzyme were present, as in the recent Cryo-EM structures (1,2), one of which contains moxifloxacin.

1. Michalczyk, E., Pabiś, M., Heddle, J. and Ghilarov, D. (2024) Structure of Escherichia coli DNA gyrase with chirally wrapped DNA supports ratchet-and-pawl mechanism for an ATP-powered supercoiling motor. bioRxiv, doi: <https://doi.org/10.1101/2024.04.12.589215>.

2. Vayssieres, M., Marechal, N., Yun, L., Lopez Duran, B., Murugasamy, N.K., Fogg, J.M., Zechiedrich, L., Nadal, M. and Lamour, V. (2024) Structural basis of DNA crossover capture by Escherichia coli DNA gyrase. Science, 384, 227-232.

Attempts to obtain very high-resolution X-ray crystal structures of holoenzyme cleavage complexes have to date been unsuccessful. However, the topo IV ParE30-ParC55 core complex retains all the key features of DNA cleavage observed for the holoenzyme. Crucially, it is amenable to crystallization and here has provided an unprecedented 2.0 Å structure. On the other hand, lower resolution cryoEM structures are now appearing of pre-cleavage and post-cleavage holoenzyme complexes of gyrase likewise revealing the presence only of a single Mg²⁺ ion per DNA strand. Given its mechanistic interest we have included the PNAS sister paper by Michalczyk et al as new reference 55.

ii. There are quite a lot of figures in the main text and I wonder if some of these should be moved to the Supplementary Materials.

The number of figures reflects the multidisciplinary approach of the work including biochemistry, quantum mechanical calculations, structural, medical and resistance aspects. We moved what figures we could to the Supplementary section without interrupting reader focus.

iii. P. 13, line 435: (26) is the wrong reference here.

Thank you. Corrected to read reference 29.

iv. Although ms is well-written there are a few places where hyphens are missing (e.g. 2.4-Å resolution), these would improve clarity.

Probably correct but not commonly used, and so we have retained the previous text.

Manuscript NCOMMS-24-40126A

'Structural basis of topoisomerase targeting by delafloxacin, a novel anionic fluoroquinolone'

Authors: Najmudin S et al

Response to Reviewers' comments

(Changes in the revised manuscript are shown in blue font).

REVIEWER COMMENTS

Reviewer #1 (Remarks to the Author): Point-by-point response:

The authors' responses to my suggestions are adequate, and I have no further concerns. I recommend the publication of this manuscript.

Reviewer #2 (Remarks to the Author):

Prior comments/concerns on the original draft have been sufficiently and satisfactorily addressed in this revised manuscript.

Reviewer #3 (Remarks to the Author):

The authors have addressed my comments satisfactorily.

(No author response required here as Reviewers 1, 2 and 3 have all signed off on the manuscript).

Reviewer #4 (Remarks to the Author):

My area of expertise related to this paper is only on the quantum mechanical calculations. Therefore, I have only a few questions regarding the computational details:

The authors stated that: The molecular framework of delafloxacin was constructed using GaussView6 and minimised first using PM7 with water as solvent. This was followed by running Gaussian 16 using the wB97XD/ Def2tzvpp basis set and then using a wB97XD/Def2-SVPP/SCRF=water level basis set.

The computational details are not correctly described.

Yes, the Reviewer is correct. In fact, a fuller description with this information was included in our original manuscript submission. However, during revision and

resubmission, we inadvertently spliced in the wrong version of the 'Computational analysis...' section (the last item in the Materials and Methods section) i.e. the text that Reviewer 4 cites above in the 2nd paragraph. We apologise for this oversight and have now replaced this section with the corrected version below:

Computational chemistry methods are now widely used in the pharmaceutical industry and elsewhere as a reliable means of small molecule structure prediction often in lieu of X-ray structure determination. In the absence of an X-ray structure of delafloxacin, we used state-of-the-art quantum chemistry methods to predict its structure with high confidence. The molecular framework of delafloxacin was constructed using GaussView 6 and minimised first using the rapid semi-empirical PM7 method with water as a continuum solvent model⁴⁷. This was followed by re-optimisation using the Gaussian 16 program and employing the ω B97X-D density functional and the double- ζ Def2-SVP basis set and then finally raising the level to the Def2-TZVPP triple- ζ basis set^{48,49} with an SCRF=water level solvation method. The resulting structure revealed a tilted N-1 aromatic ring in accord with that seen in the free drug X-ray structures of a closely related delafloxacin derivative (identical except for a slightly modified N-7 azetidiny substituent) and of the structurally related fluoroquinolones trovafloxacin and tosufloxacin^{46, 62}. The predicted delafloxacin structure closely overlaps the drug conformation seen here in the topo IV cleavage complex.

Inclusion of this text now addresses the four points raised by the Reviewer (listed below) which are based on our previous incorrect Computational section.

- 1) First, what the authors refer with running Gaussian16? Did they perform a geometry optimization?
- 2) Second, ω B97XD/ Def2tzvpp is a level of theory not just a basis set. The ω B97XD is the DFT functional used and the Def2tzvpp is the basis set.
- 3) The second part, "then using a ω B97XD/Def2-SVPP/SCRF=water level basis set." is also not properly explained. This calculation is a single point energy calculation based on the optimized structure?
- 4) The ω B97XD/Def2-SVPP/SCRF=water is again a level of theory, not a "level basis set."

I can not recommend the paper for publication without these issues being addressed.

We believe these points are all covered fully in the newly revised manuscript.

We should like to thank all the Reviewers for their helpful and insightful comments.

Finally, we have made a small number of clarifying changes to the Main text and Supplementary Material that address minor points of information or accuracy, viz:

-Results: section on ‘Canonical features of the delafloxacin-stabilized complex’. Second sentence corrected to state ‘with the dimer as the asymmetric unit’.

-Results: section on ‘Delafloxacin binds in a tilted ring conformation’. Corrected text to read ‘In the absence of a crystal structure of delafloxacin, we used computational chemistry methods (PM7 and Gaussian 16, see Methods section) to calculate the structure of the drug.

-Results: section on ‘A ‘water-metal ion bridge’ stabilizes the intercalated drug: role in resistance’. ‘drug C-4 carboxyl group’ corrected to read ‘drug C-3 carboxyl group’

-Discussion, last line of paragraph 8 and Supplementary Fig 7 legend. In both, corrected ‘nucleotides 21-22’ to read ‘20-21’.

-Discussion: paragraph 10. Changed sentence beginning ‘Subsequently, from a 2.4 Å structure..’ to read ‘Subsequently, from a 2.4 Å E-site structure.’

-Materials and Methods: section on Crystallization, 2nd paragraph. Corrected fourth sentence to clarify that the same reservoir solution (particular to each set of conditions) was used in forming crystals with and without nucleant.